



# Re-evaluating safety risks of multifunctional dikes with a probabilistic risk framework

Richard Marijnissen[1], Matthijs Kok[2], Carolien Kroeze[1], Jantsje van Loon-Steensma[1,2]

[1]Water Systems and Global Change group, Wageningen University & Research, Wageningen, P.O. Box 47 6700 AA
Wageningen, the Netherlands
[2]Hydraulic Engineering, Delft University of Technology, P.O. Box 5048, 2600 GA Delft, the Netherlands

*Correspondence to*: R.J.C. Marijnissen (richard.marijnissen@wur.nl)

**Abstract.** Multifunctional use of flood defences is often seen as a disadvantage for flood protection. Safety assessments of multifunctional dikes only require functions do not negatively affect safety but leave potential synergies untapped. This study synthesizes the new probabilistic approaches to evaluate the safety of multifunctional flood defences employed in the Netherlands after the introduction of the new Water Act and explores how this affects the assessed risk of flooding. While a traditional conservative approach does lead to safe assessments, a probabilistic approach assesses a higher protection level of the dike. Positive contributions of functions to safety can be included in a probabilistic approach even when in a critical state there is a negative contribution to safety. In a probabilistic approach the probability of such scenarios is made explicit. Multifunctional flood defences thereby are more safe than is expected from conservative assessments only.

## 1 Introduction

### 1.1 Evolution of the flood risk approach

With sea-level rising globally and an expected rise in extreme rainfall events due to climate change the risk of floods is increasing (Bouwer et al., 2010;Hirabayashi et al., 2013). In order to develop sufficiently strong infrastructure to prevent these catastrophes a framework  is needed to assess the safety the infrastructure provides. Risk based approaches towards flood protection have been performed all over the world to inform decision makers on effective flood risk measures in spite of the large uncertainties (Jonkman et al., 2009;Kheradmand et al., 2018;Hall et al., 2003). Nevertheless, a better the understanding of the fragility of flood protection measures, including innovative ones like natural flood defences (Temmerman et al., 2013), is instrumental to properly evaluate the flood risk in the future.

This is true for the Netherlands especially where about 60% of its area is already prone to flooding from the sea or rivers (Slomp, 2012). Flood protection has always been a priority yet standards were only formalised in the past century. The first Delta committee, established to advise the government on flood risk after the large flood of 1953, advised to set a design water level with an acceptably small exceedance probability that flood defences need to retain. The acceptable exceedance probability followed from an economic optimisation between investment costs and obtained risk reduction (Delta Committee, 1960).  This approach was the basis of the Water Act which sets the required protection level of all dikes in the Netherlands. As of January 2017 a new probabilistic approach has been adopted in the Water Act to which flood defences need to comply by 2050 (Ministerie van Infrastructuur en Milieu, 2016a). Because of the shortcomings of the old approach, economic developments and growing concerns over climate change the second Delta Committee advised to revise the water-level exceedance based risk approach into a full probabilistic approach (Delta Committee, 2008) and the change was made in 2017.

The metric for the protection level of flood defences used to be the most extreme event with a specified exceedance probability it is still able to retain (Van der Most et al., 2014). Many studies have argued for a comprehensive probabilistic approach towards assessing the protection level provided by flood defences before (Apel et al., 2006;Vrijling, 2001;Hall et al., 2003).



The Dutch Water Act is the first to put these principles into practise on a nation-wide scale. While these approaches were developed for dikes that serve flood protection only, in practise many dikes have features serving other functions than flood protection. It is yet unclear how such functions must be included in probabilistic safety assessments.

### 1.2 Multifunctional flood defences

Multifunctional flood defences (MFFDs) are structures designed for the purpose of flood protection while simultaneously enabling other uses (Voorendt, 2017). Combining dikes with other functions is fairly common. Dikes can have roads on top, cables and/or pipelines running through them, structures on them, or are part of a historic landscape. In the Netherlands alone a majority of dike reinforcement projects already face the presence of one or multiple additional functions. Usually, enabling multiple functions requires strengthening of the dike beyond the minimal requirements for a traditional dike to account for

uncertainties related to the functions (van Loon-Steensma and Vellinga, 2014). Other functions do not need to be a detriment to safety. For example, the development of nature for flood protection services is an attractive option for future climate adaption (van Loon-Steensma et al., 2014) as such flood defences with green foreshores can reduce the risk of flooding by natural processes (van Loon-Steensma and Kok, 2016;van Loon-Steensma et al., 2016).

Flood defences can strengthen other values when functions are properly integrated (Lenders et al., 1999;van Loon-Steensma et al., 2014).  In urban areas where space is limited there is continuous pressure to build on or integrate structures with the flood defence (Stalenberg, 2013). In rural areas nature-based solutions have gained interest, which combine beneficial properties of natural systems for flood protection (e.g. wave attenuation by vegetation on foreshores) with conserving or developing important natural values (Temmerman et al., 2013;Pontee et al., 2016). Especially in the Netherlands these

developments favour the implementation of a multifunctional flood defence due to the limited space and government policy to consider other uses (e.g. the natural, historical, economical, etc.) (van Loon-Steensma and Vellinga, 2014).

Despite the large number of multifunctional dikes and incentives the tools to assess the safety of MFFDs have still been limited to rules of thumb on the one hand and in-depth studies on the other. Unless the multifunctional aspect is perceived to be of

sufficient importance to justify a tailor-made study, assessments are often limited to showing other functions do not significantly diminish the safety of the flood defence while ignoring potential positive contributions to safety. Using a conservative approach for dike assessments where functions can only negatively influence flood risk  does ensure safe dikes from a flood risk perspective but may hamper the implementation of efficient multifunctional dikes by requiring larger or more expensive dikes.

### 1.3 Aim

There is a need for improved flood defences due to climate change (rising sea-levels, higher river discharges) and socio-economic developments. The number of people exposed to a high risk of flooding is expected to increase from 271 million in 2010 to 345 million in 2050 due to socio-economic growth alone (Jongman, Ward, & Aerts, 2012). By 2100, 168 million people per year will experience floods due to sea-level rise. By reinforcing dikes this number can already be reduced by a

factor of 461 (Hinkel, van Vuuren, Nicholls, & Klein, 2013). While reinforcing dike systems, there is plenty of opportunity to combine multiple functions with dikes.

However, the means to determine the safety provided by multifunctional flood defences remain limited to conservative approaches where functions can only be shown to have no significant negative influence. Spurred by the threat of increasing

flood risks by climate change and the revised legislation on flood standards in the Netherlands a new probabilistic framework to assess multifunctional flood defences is emerging that can be used for a wider context. The aim of this paper is to synthesize



the new approaches to evaluate the safety of MFFDs employed in the Netherlands into a single cohesive framework and evaluate how this new probabilistic approach towards MFFDs can change the assessed safety compared to the commonly applied conservative approach towards MFFDs .

To this end, we first analyse the existing official framework for assessing multifunctional dikes in the Netherlands and explore alternative frameworks in both scientific and grey literature for a probabilistic risk-based approach towards assessing MFFDs as required by the new Water Act. These are synthesized in an adapted framework (section.2). Secondly we explain the methods used to calculate the probability of failure of several dikes using the synthesized probabilistic approach and the traditional conservative approach (section 3) to show the differences in assessed safety level (section 4). Finally we discuss
the implications and results (sections 5 and 6). By illustrating how a probabilistic approach towards multifunctional use can affect the assessed level of safety, new types of integrated solutions can be more fairly compared to traditional dikes both in the Netherlands and outside.

## 2.        Formulating a framework for MFFD assessment

### 2.1 Official Dutch guidelines for MFFD dike assessments and design

The methods to assess flood defences in compliance with the official Dutch safety standard are communicated in official guidelines (Ministerie van Infrastructuur en Milieu, 2016b;Ministry of Traffic and Water Management, 2007;Rijkswaterstaat, 2017). The assessment can be performed on different levels: basic, detailed, and tailored. Basic assessments are a quick-scan with simple rules to exclude flood defences with an insignificantly low failure probability. Detailed assessments consist of design formulas and models taken or adapted from Dutch design manuals and are commonly applied for (initial) designs and
assessments. These are suitable for predicting the failure of dikes where general descriptions of dike failures can be applied. Such generalisations are not always suitable for MFFDs. Tailored assessments allow for the use of advanced models and experiments outside the guidelines to assess the probability of failure as accurately as possible. These assessments require a large amount of information for a specific location and are generally expensive to perform. The dike needs to pass at least one of these assessments to be considered safe and a proper design ensures the dike will pass the assessments for its entire designed
lifespan.

In the official Dutch framework, multifunctional use of the dike is considered either directly as objects on the dike, by the materials used, or indirectly by the geometry of the dike. When only the geometry of the dike is affected or a different material is used (e.g. to integrate with the surrounding landscape) the official framework can still be applied (Slomp et al., 2016).
However, if the function is facilitated by a Non-Water retaining Object (NWO), e.g. a house or pipeline, an additional assessment must be made for the NWO. For a few functions a basic safety assessment is described in guidelines (structures, vegetation and traffic) (Deltares, 2012;STOWA, 2000;TAW, 1994, 1985;STOWA, 2010;Rijkswaterstaat, 2017). Only for pipelines a more detailed assessment is available following the Eurocode (NEN, 2012) which ensures the pipeline itself has an acceptably small probability of failure. If a dike cannot be approved by a basic assessment and no suitable detailed assessment
is available, a tailored assessment for that specific dike section with NWOs must be made.

The philosophy of a basic assessment is to rule out the possibility of the NWO affecting the dike significantly. Hence, the dike is considered safe only if the dike is dimensioned such that the zone of influence of the NWO does not extend into the minimum dike profile needed to meet the safety standard (see Fig. 1). As a result, in basic assessments the NWO is always
assumed to be in its most critical state during design conditions (e.g. uprooting of a tree). This is the conservative approach to assessing the influence of functions on the safety because the probability of multifunctional elements being in a critical state





is not considered. The ambition of the Dutch Water Act is to consider the actual probability of flooding which necessitates a risk-based approach to these elements.

**2.2 Synthesizing a risk-based approach to MFFD design**

The scientific basis for the risk-based framework adopted in the Netherlands was presented by Vrijling (2001). The risk of a

flood is decomposed into a fault tree of failure mechanisms, each of which can be described with a limit state function and evaluated probabilistically. Limit states are common for designing structures in Civil Engineering and define when a structure collapses resulting in damages and casualties (ultimate limit state) or can no longer perform its intended use (serviceability limit state) (Gulvanessian, 2009). Vrijling's approach of structuring the ultimate limit states of flood defences into a fault tree for risk analyses has been incorporated in many frameworks of flood defences, e.g. (Steenbergen et al., 2004;van Gelder et al.,

2008;Slomp et al., 2016), and has already been applied on a large scale to evaluate the Dutch flood defences (Jongejan et al., 2013). However, the framework was developed for monofunctional flood defences.

Studies on MFFDs specifically are available. However, the developed frameworks address different aspects like: to identify the degree of spatial and structural integration (Ellen et al., 2011b;Voorendt, 2017;Van Veelen et al., 2015), to identify costs

and benefits (Anvarifar et al., 2013), to identify the threats and opportunities of functions (Anvarifar et al., 2017), and to identify and evaluate flexibility for MFFDs (Anvarifar et al., 2016). Other studies on MFFDs tend to only focus on the effects of a specific function or failure mechanism (Chen et al., 2017;Bomers et al., 2018;Zanetti et al., 2011). Only recently an assessment framework specifically for hybrid nature-based flood defences was put forward accounting for multiple failures by putting vegetation-specific equations directly into the assessment procedure (Vuik et al., 2018).

Pending an official framework practitioners in the Netherlands have used approaches to integrate multifunctional dike elements. One such approach was put forward for trees through the use of scenarios such as uprooting (Deltares, 2012) but it was left unclear how these failure probabilities can be implemented in the overall framework (Witteveen+Bos, 2013). An approach for assessing NWOs as indirect failure mechanisms with scenarios is being suggested in these cases (Knoeff, 2017).


Formulating a practical framework for the assessments of MFFDs is challenging due to the large variety of possible configurations and range of functions. While in scientific literature decision frameworks and the knowledge gaps of specific functions and failures are addressed, the assessment framework of the WBI2017 addresses how to evaluate the overall safety of the dike system but lacks the means to evaluate the additional functions. Through scenarios the inclusion of unspecified

functions can be evaluated in different states through simple or complex models in literature while preserving the established structure of the existing Dutch framework. Scenarios in this context are different possible states of a function with a probability of occurrence in which the function affects the flood defence. By assessing each scenario and weighing the probability of failure in each scenario by the probability of the scenario, the probability of failure of the flood defence is calculated accounting for the uncertainty in the state of the function. We therefore synthesize the methods for MFFD assessments in the Netherlands

in Fig. 2.

- Step 1: Establish the required safety level of the dike segment
- Step 2: Assign a portion of the required safety level to unknown/unquantifiable risks
- Step 3: Distribute the remaining failure budget across the known failure mechanisms
- Step 4: Divide the dike in (close to) homogeneous sections
- Step 5: Determine a representative cross section and safety level taking variations along the dike section into account (length effect)




- Step 6 (Addition): Determine the scenarios, i.e. states in which the NWO affects the flood defence differently, assess the probability of these scenarios, and combine them based on their probability of occurrence.

The difference between a basic approach and the risk-based approach is the addition of step 6. In a basic approach, i.e. a detailed assessment without NWOs followed by a basic NWO assessment to exclude significant potential negative influences, first a dike cross-section would be designed with the criteria found in steps 1 to 4 and then adapted such that the influence of the intended NWO is outside the designed profile. In the risk approach the effects of NWOs should be calculated directly with the scenarios in step 6 and combined with their probability of occurrence to arrive at a safe cross-section.

## 3 Application of the risk frameworks

### 3.1 Comparing the basic framework with the expanded risk-based framework

To answer how a more probabilistic approach towards multifunctional dikes can affect the evaluated safety compared to a monofunctional dike, we assess a set of MFFDs with the new probabilistic approach and a the traditional conservative approach (see Table 1). The calculations are performed on a cross-sectional level. The reliability of a cross-section is calculated for the most common dike failure mechanisms by probabilistically evaluating the limit state functions for the different scenarios. To combine the different failure probabilities the fragility curves of the mechanisms can be used (Bachmann et al., 2013) to arrive at the probability of failure.

### 3.2 Failure mechanisms

To assess the risk of a flood it is important to know the mechanisms by which the flood defence could fail. Though many failure mechanisms are possible (Kok et al., 2016) the vast majority of documented dike failures worldwide (Danka and Zhang, 2015) are the result of 3 dominant mechanisms: overtopping (resulting in erosion of the inner slope), internal erosion (also referred to as piping), and inner slope stability. Within the Netherlands predominantly overtopping and slope instability have been the cause of dike breaches in the past (Van Baars and Van Kempen, 2009). For this study the probability of a flood is calculated by considering the failure mechanisms overtopping, piping, and macro stability (see Table 2). Whether the flood defence fails by a failure mechanism is expressed in a limit state function:

$$Z = R - S \tag{3.1}$$

where Z<0 denotes failure, R is the resistance to failure, and S is the soliciting load.

For overtopping and overflow the load (S) is the amount water flowing over the dike while the resistance (R) is the capacity of the crest and inner slope to resist the flow of water without eroding. For piping the method of Sellmeijer et al. (2011) is used to calculate the stability of the sand particles in the subsoil under a pore water pressure gradient. It is expressed as a critical head difference (R) that cannot be exceeded by the head difference across the dike (S). Macro stability is calculated within the program D-Geo Stability (Deltares, 2016) with the stability method by Van (2001) and ground water model by TAW (2004). The method by Van (2001), like the Bishop (1955) method, calculates the sum of the driving moments (S) and the total resisting moment (R) along the slip plane. However, it also accounts for uplift forces on the interface of aquifers present beneath most dikes. The resulting limit states are:

$$Z_{\text{overflow \& overtopping}} = q_c - q \tag{3.2}$$
$$Z_{\text{piping}} = H_c - H \tag{3.3}$$
$$Z_{\text{macro stability}} = \Sigma M_R - \Sigma M_S \tag{3.4}$$

Here $q_c$ is the empirically determined critical overtopping discharge, q is the overtopping discharge calculated according the methods of van der Meer et al. (2016) and TAW (2002), $H_c$ is the critical hydraulic head according to Sellmeijer et al. (2011),





H is the difference in water level in front and behind the dike, $\Sigma M_S$ is the sum of the active moments in the critical slip plane, and $\Sigma M_R$ is the sum of resisting moment in the critical slip plane.

### 3.3 Probabilistic procedure

Multiple procedures are available for calculating the reliability of a flood defence. A fully probabilistic procedure like Monte

Carlo relies on evaluating the limit state function for many variations of the random variables and determines the failure probability as the number of failures over the total number of samples. Meanwhile, a semi-probabilistic approach evaluates the limit state function once and captures uncertainties with (partial) safety factors to determine (non)failure. A probabilistic procedure like the first order reliability method (FORM) iteratively converges to an approximation of the probability of failure (Hasofer and Lind). This option was chosen as it does not require millions of evaluations of the limit state function to find the

low failure probabilities required for dikes while still retaining the probabilistic distribution of the variables otherwise lost in a semi-probabilistic approach.

While the FORM procedure can approximate the failure probability of a single limit state function of a single failure mechanism, a combination of failure mechanisms is more complex to evaluate. When the only dependence between failure

mechanisms is assumed to be the water level, each failure mechanism becomes an independent event for each discrete water level such that the probability of failure of the system is:

$$P_{f,\text{sys}|h} = P_{sys}(f|h) = 1 - \prod_{i=1}^{n}\left(1 - P_{f,i|h}\right) \qquad (3.5)$$

Where $P_{f,i|h}$ is the probability of failure given water level h for the $i^{th}$ failure mechanism and $P_{f,\text{sys}|h}$ is the probability of failure given water level h. Repeating this calculation across all water levels results in the fragility curve of the system to the water level (Bachmann et al., 2013). The failure probability of the system is computed by integrating the fragility curve of the system

($F_R(h)$) over the probability density function (PDF) of the water level ($f_h(h)$):

$$P_{f,\text{sys}} = \int_{h=-\infty}^{h=\infty} f_h(h) * F_R(h)\, dh \qquad (3.6)$$

Eq. 3.6 is discretised to:

$$P_{f,\text{sys}} = \sum_{j=1}^{m} P(h_j) * P_{sys}(f|h_j) \qquad (3.7)$$

Low failure probabilities can more easily be expressed in terms of the reliability index which is defined as:

$$\beta = -\Phi^{-1}(P_f) \qquad (3.8)$$

Where $\Phi^{-1}$ is the inverse standard normal cumulative distribution function.

The probabilistic procedure described above has been utilised before successfully by Lendering et al. (2018) and Bischiniotis et al. (2018) to compute the reliability of canal levees and a cost-optimal river dike respectively. An overview of the entire process as applied in this study is schematised in Fig. 3.

### 3.4 Case-study

#### 3.4.1 Setting and cross sections

To test how a risk approach can affect the calculated level of safety 8 cross-sections of multifunctional dike profiles (Fig. 4) are evaluated with a conservative, probabilistic and monofunctional approach (see Sect. 3.1). Each profile represents a common reinforcement strategy. The dike is situated in a riverine area where both the flood plain and hinterland are occupied by a function, nature and a structure respectively.





Each function can potentially damage part of the dike section. For the purpose of this study the functions have been simplified so these can be incorporated directly in variables of the limit state functions or dike geometry (see Sect. 3.4.2). In alternatives 1 and 5 the hinterland function remains separate from the dike itself, while in the other alternatives the structure becomes an integral part of the flood defence. By reviewing the options we explore how the safety after the reinforcements is evaluated in

each framework.

### 3.4.2 Schematisation of the functions

The schematisation of functions in this study has been based on the fact-sheet by Knoeff (2017) for incorporating indirect failure mechanisms in assessments. For each mechanism scenarios are defined in which the object (e.g. tree, structure, pipeline, etc.) affects the failure mechanisms. The probability of failure can then be calculated for each scenario. The total probability

of failure for the specific mechanism can be computed be weighing the probability of failure of each scenario with the probability of the scenario.

A natural flood plain can add ecological, landscape and recreational values to the flood protection system. However it comes with implications for safety. Woody vegetation can penetrate the clay top soil resulting in cavities within the clay (Zanetti et

al., 2011). This will allow water to seep into the aquifer closer to the dike increasing the risk of piping. In greater densities woody vegetation like willows can be beneficial to safety by damping incoming waves (de Oude et al., 2010).

For the examples in this study it is assumed woody vegetation develops somewhere in the representative cross-section. Following the conservative estimation by TAW (1994) it has a 2% annual probability of failure evenly distributed along the

foreshore of the dike. As the density of trees is too small for significant wave damping this effect is ignored. If the vegetation has disturbed the top soil the effective length for piping was reduced to the distance between the dike outer toe and the location of the disturbance.

Within the base profile there is a structure 3 m behind the inner dike toe. The structure is taken to be 15 m wide, exerts a weight

of 17 kN/m and is embedded 1 m into the soil with on a shallow foundation without additional geotechnical measures like piles or sheet pile walls . The position of the structure remains fixed for each reinforcement strategy except for the robust dike where the structure is raised onto the slope. The structure is only considered in two 2 states: present in which the load is exerted, or absent in which case the load of the structure is absent and a hole is present in the profile at its location. When the structure is present an 'open' grass cover is assumed as along the edges of the structure the grass will not be present. When the structure

is absent the large stretch of bare soil will be vulnerable. It is assumed that in this situation only an insignificant amount of overtopping (q<0.1 l/m/s) is acceptable (see Table 3). The probability of the structure being absent is taken to be 1% as a conservative estimate.

### 4.   Results

The summary results are presented in Fig. 5. As expected the conservative approach consistently yields the highest probabilities

of failure for the assessed dikes. The probabilistic assessment of the functions and the monofunctional assessment yield a lower probability of failure. Whether a function has a net positive or negative influence on the safety of the dike becomes only apparent by comparing these.

The weight of the structure has a noticeable net positive influence on the reliability when it is included as part of a reinforcement

(see profiles 2, 3 and 5) which is lost in a conservative assessment. Meanwhile it is also clear that the weight of the structure





is less beneficial in a different configuration (profile 4). With a berm as both the structure and berm add weight, but the structure introduces a risk of the berm being lost when the structure fails. The effect is also limited when the structure remains just outside of the profile (profiles 0, 1, 2) or stability as a failure mechanism is not significantly contributing to the probability of failure (profile 7).

While in the calculations with a structure the clay cover on the flood plain is assumed to be intact along the full length, including uncertainty because of unmanaged activity on the flood plain (nature, recreation) has a large effect on piping failure. This is especially true in the conservative approach where the entire length of the flood plain is not taken into account in the assessment where it leads to a different perception in the need for piping specific reinforcement measures. There is an increasing

discrepancy between the conservative assessment and the other assessments mainly due to very different assessments of the risk of piping.

Finally the difference in probability of failure between a monofunctional dike and a multifunctional dike depends on the reliability of the monofunctional dike itself. Unless there are large differences in the schematisation of a failure mechanism

(as was discussed for piping), differences in failure probabilities between assessments scale roughly by the same order of magnitude as the decrease in failure probability after a reinforcement (Fig. 5 note the log-scale for the probability of failure). However, the relative differences become more pronounced leading to proportionally higher failure probabilities in a conservative assessment compared to a probabilistic assessment.

## 5. Discussion

The results show a large difference between the reliability assessed between the conservative approach and the probabilistic approach. A prevailing view against multifunctional use of flood defences is that these require larger dimensions to meet the same safety standard as a traditional dike (Ellen et al., 2011a;van Loon-Steensma and Vellinga, 2014). However, as the case-study above illustrated this perception only holds true for a conservative approach that only assesses the parts of the dike unaffected by other functions. With a more probabilistic approach towards additional functions the perceived negative

influence of the functions was significantly smaller or could even result in a net positive influence. Positive contributions of the function under likely conditions can be included as well as the likelihood of the function affecting the flood defence negatively.

A drawback of the probabilistic approach is that it needs specific information about the risks and states of the functions before

an assessment can be conducted. For example, erosion around or over discontinuities during overtopping (possibly due the presence of multi-functional elements like a road) is highly variable and hard to capture in a generic limit-state function even with well-calibrated models (Hoffmans et al., 2009;Bomers et al., 2018). Depending on the sensitivity of the failure probability to these processes assumptions on effects and statistical distributions would need to be increasingly conservative to guarantee the safety level is met. However, new information on the risks of functions is becoming increasingly available through ongoing

research (Aguilar-López et al., 2018;Vuik et al., 2018). Furthermore, new techniques are being employed to continuously monitor the dikes in detail (Hanssen and Van Leijen, 2008;Herle et al., 2016) while advances in remote sensing allow for closer monitoring of the state of foreshores (Niedermeier et al., 2005;Friess et al., 2012). As a result, a probabilistic approach towards functions can capitalise on these advances by updating the previously assumed risks in assessments with observations of the actual performance of MFFDs over time.




Aside from the effects of functions themselves, other uncertainties influence how much other functions can affect the level of safety. For piping Aguilar-López et al. (2015) demonstrated that reducing the uncertainty in the seepage of the soil of a multifunctional dike by correlating grain-size and hydraulic conductivity the probability of a piping failure is already reduced. Lanzafame (2017) concluded variability introduced by vegetation has only a small effect on the probability of a slope failure

due to larger uncertainties in strength and seepage of the ground. In contrast a relatively small disturbance by burrowing animals in a fragile dike has resulted in a breach under conditions it had previously survived (Orlandini et al., 2015). The influence of uncertainties was also observed within this study through the case of a robust dike. As the reliability of the dike itself increases, the influence of a function on the level of safety decreases as the added variability of the function becomes smaller compared to the uncertainties in other parameters the dike was already designed for. This effect of dike reliability on

the influence of functions has implications. An increase in failure probability due to functions is likely to be over-estimated in a traditional assessment for dikes with a high protection level while similarly for these dikes also only a limited decrease in failure probability can be expected from beneficial functions. Conversely, dikes with a low protection level are influenced more by both beneficial and detrimental effects of additional functions.

This study only looked at the effects of functions on flood protection. However, these functions can have their own set of requirements that should be taken into account. For example structures need to comply with building codes, flood protection in nature reserves is subject to environmental protection regulations while to preserve landscape values substantial dike heightening may be unacceptable. How much such additional non-flood protection requirements influence the design of dikes should be researched for a successful implementation of MFFDs.

While the current study looked at assessments for an existing situation, relating and managing uncertainties of functions to uncertainties in future climate conditions will be crucial for a probabilistic application of additional functions in designs. Predictions for future sea-level rise in the coming century vary between 0.23 and 0.98 m (IPCC, 2013). Incorporating beneficial multi-functional uses of flood defences, either natural like marshes or man-made like structures, can become an asset to achieve

the levels of flood protection needed in the future.

## 6.   Conclusion

We analysed how a full probabilistic approach towards multifunctional flood defences can change the assessed safety compared to the commonly applied conservative approach where additional functions can only be shown to have no significant negative influence. Although a probabilistic assessment was not forbidden, new regulations and insights of the Water Act in

the Netherlands stimulate a probabilistic assessment of flood protection. Therefore a framework incorporating multifunctional elements probabilistically was synthesized. The overall conclusion is that application of a probabilistic approach towards additional functions will lead to a lower assessed risk of flooding because: 1) positive contributions of functions to safety can be included, even when in a critical state there is a negative contribution to safety and 2) the risk of a functions being in such a critical state is made explicit. Another important aspect is that effects of functions on safety become smaller as the reliability

of the dike increases. Therefore monofunctional dikes with already high protection levels are more suitable to be combined with functions detrimental to safety whereas dikes with low protection levels can benefit more from functions that contribute to safety.

Based on the results we recommend that a probabilistic framework is further developed and implemented for including

multifunctional elements into dike assessments. While many knowledge gaps are still present in quantifying the effects of functions, incorporating scenarios in which a function can harm or help flood protection can already provide insights in



synergies that can be exploited or dangers that can be mitigated. Furthermore it is expected that with the growing number of methods to monitor dike performance and ongoing studies in dike failures these gaps can be filled in the future.

To this end further research is required into monitoring schemes that can be used to improve future assessments of the functions
on the dike. Additionally, more research is needed to assess how multifunctional elements influence the safety of dikes over longer periods especially in relation to the large uncertainties involved in climate change.  A real-world case-study for design should be used to explore how these aspects can be incorporated in practise.

### 7.  Author contributions

The study and methodology were conceived by RM, JvL and MK. RM carried out the analyses, produced the results and wrote
the manuscript under the supervision of JvL, MK and CK. The results were discussed and reviewed among all authors.

### 8.  Competing interests
The authors declare that they have no conflict of interest.

### 9.  Acknowledgements

This work is part of the Perspectief research programme All-Risk with project number P15-21, which is financed by NWO Domain Applied and Engineering Sciences. We thank Wim Kanning for his advice in the probabilistic stability calculations. Furthermore we would like to thank Harry Schelfhout and Reindert Stellingwerf for the discussions on the current practices for multifunctional dikes.



**Appendix**

**A: Case-study parameters**

The dike geometry of the base case is captured by the variables in Table A1.

The soil was divided into 3 layers: the dike core, the blanket layer and the aquifer. Representative values for the soil layers were taken from known soil types in the Dutch riverine area (Table A2, Table A3 and Table A4).

Hydraulic load parameters are given in Table A5. Representative water and wind characteristics were estimated from the

hydraulic loads database of the upper Rhine area in the Netherlands which is available as part of the WBI software. For simplification the wind direction is only considered in the direction perpendicular to the dike.

**B: Overflow and overtopping limit state function**

Overflow is calculated directly from the water level (h) and crest height ($z_{crest}$) by the formula for a broad crested weir:

$$q_{overflow} = \sqrt{2g} * \frac{2\sqrt{3}}{9}(h - z_{crest})^{\frac{3}{2}}$$  (B1)

To calculate the overtopping discharge first the significant wave height ($H_s$) and period ($T_s$) perpendicular to the dike are

estimated from the water depth (h), fetch length (F), and wind speed ($u_{wind}$) with the equations of Bretschneider (1957) as presented by Holthuijsen (1980):

$$F_x = \frac{gF}{u_{wind}^2}$$  (B2)

$$h_x = \frac{gh}{u_{wind}^2}$$  (B3)

$$p_1 = \tanh(0.53 * h_x^{0.75})$$  (B4)

$$p_2 = \tanh(0.833 * h_x^{0.375})$$  (B5)

$$H_s = 0.283 * \frac{u_{wind}^2}{g} * p_1 * \tanh\left(0.0125 * \frac{F_x^{0.42}}{p_1}\right) * m_{Bret,H}$$  (B6)

$$T_s = 7.54 * \frac{u_{wind}}{g} * p_2 * \tanh\left(0.077 * \frac{F_x^{0.25}}{p_2}\right) * m_{Bret,T}$$  (B7)

With the wave characteristics the average overtopping discharge is calculated following the formulas by TAW (2002) and van der Meer et al. (2016). Since no berm is present on the dike of the case-study and waves are assumed perpendicular factors

related to these aspects are omitted.

$$q_1 = \min \left( \begin{array}{c} \frac{0.067}{\sqrt{\tan \alpha_{out}}} * \xi_0 * \exp\left(c_1 * \frac{z_{crest} - h}{H_s} * \frac{1}{\xi_0 * \gamma_f}\right) \\ 0.2 * \exp\left(-2.6 * \frac{z_{crest} - h}{H_s} * \frac{1}{\gamma_f}\right) \end{array} \right) * \sqrt{g * H_s^3}$$  (B8)

$$q_2 = 10^{c_2} * \exp\left(-\frac{z_{crest} - h}{\gamma_f * H_s * (0.33 + 0.022 * \xi_0)}\right) * \sqrt{g * H_s^3}$$  (B9)

$$q_{overtopping} = \begin{cases} q_1 & \xi_0 < 5 \\ 10^{\frac{\log(q_1) + \log(q_2)}{2}} & 5 \geq \xi_0 \geq 7 \\ q_2 & \xi_0 > 7 \end{cases}$$  (B10)

A description and values for the variables are presented in Table B1.

The limit state function is then evaluated as:



$$Z_{\text{overflow and overtopping}} = q_c - q_{\text{overflow}} - q_{\text{overtop}} \tag{B11}$$

### C: Piping limit state function

Piping is evaluated with the ground water schematisation of TAW (2004) and piping erosion formulae of Sellmeijer et al. (2011). To simplify the calculation these assumptions are made: a finite foreshore blanket is considered of a significant thickness (d > 1 m) and impermeable (k <1*10⁻⁷ m/s), the hinterland blanket is significant and continuous in, there is no flow

5 of water through the aquifer from other sources than the river, and finally the blanket layer has the same properties at the foreshore and hinterland. Following these assumptions the response in water head just behind the dike during high water is determined by the leakage length ($\lambda$) and response factor (r) at the end of the leakage path with length L which is the distance from the entree point to the dike ($L_{\text{entree}}$) plus the width of the dike ($L_{\text{dike}}$).

$$\lambda = \sqrt{k_{\text{aquifer}} * d_{\text{aquifer}} * \frac{d_{\text{blanket}}}{k_{\text{blanket}}}} \tag{C1}$$

$$r = \frac{\lambda}{L + \lambda} * \exp\left(\frac{-\left(\frac{L_{\text{dike}}}{2} + L_{\text{entree}}\right)}{\lambda}\right) \tag{C2}$$

The hydraulic head difference within the aquifer layer across the dike and foreshore (H) is calculated as:

$$H = h * (1 - r) \tag{C3}$$

The critical head difference ($H_c$) is calculated with the piping erosion formulae of Sellmeijer et al. (2011):

$$F_R = \frac{\gamma_p - \gamma_w}{\gamma_w} * \eta * \tan\theta * \left(\frac{RD}{RD_m}\right)^{0.35} \tag{C4}$$

$$F_S = \frac{d_{70}}{\sqrt[3]{\kappa L}} * \left(\frac{d_{70m}}{d_{70}}\right)^{0.6} \tag{C5}$$

$$F_G = 0.91 * \left(\frac{d_{\text{aquifer}}}{L}\right)^{\frac{0.28}{\left(\frac{d_{\text{aquifer}}}{L}\right)^{2.8} - 1} + 0.04} \tag{C6}$$

$$H_c = F_R * F_S * F_G * L \tag{C7}$$

Failure occurs when the critical head level is exceeded by the head difference and the resistance of the blanket layer:

$$Z_{\text{piping}} = m_p * H_c - (H - 0.3 * d_{\text{blanket}}) \tag{C8}$$

The variables introduced by Eq. (C4) to Eq. (C8) are given in Table C1. and are based on estimates provided for WBI assessments, expect for the intrinsic permeability ($\kappa$) which is directly converted from the permeability of the aquifer ($k_{\text{aquifer}}$).

### D: Macro stability limit state function

The macro stability of the dike is evaluated using the schematisation of the phreatic surface of a clay dike from the TAW

20 (2004) following the WBI 2017 guidelines (see Fig. D1). The TAW (2004) schematisation assumes a drop in the phreatic surface on the interface of the dike with the outside water (1 m as by default) and a linear drop towards the inner toe. The water head in the aquifer was calculated the as for piping (see appendix C).

The stability of the slope is calculated with the method by Van (2001) for the slip plane and works on the same principle as

25 the method by Bishop (1955). The main difference between the methods is the separation of the slip plane in an active circle connected by a straight section followed by a passive circle. The centres of these circles of the critical slip plane ($R_A$ and $R_P$) are found iteratively using the D-stability software (Deltares, 2016).





The slip plane is divided into slices and the net force induced by each slice is calculated. If the moment induced by the active slices ($\Sigma M_S$) is greater than the combination of friction forces and moments induced by the passive slices ($\Sigma M_R$) the slope is unstable. This is both expressed in a factor of safety ($F_S$) and Z function.

$$F_S = \frac{\Sigma M_R}{\Sigma M_S} \tag{D1}$$

$$Z_{macrostability} = F_S - 1 \tag{D2}$$

To calculate the probability of failure with FORM the factor of safety needs to be evaluated during each iteration with D-stability. An experimental version of D-stability with an additional piece of software from the same developers called the probabilistic toolkit (PTK) was utilised to automatically execute D-stability with updated parameters calculated by the FORM algorithm in the PTK.

The iterative procedure of finding the critical slip plane is both computationally demanding and complicates conversion in the probabilistic FORM algorithm. To speed up the procedure in the computation first a test run is performed using average soil strength parameters at a fixed critical slip plane with a water level halfway at the crest . With the results of the first indicative run , stochastic variables with little to no influence ($|\alpha|$<0.001)  are set as constants. Then the entire model was run for each discretised water level.

After the run the fragility curve was checked for points where no convergence was achieved with FORM or a non-critical slip circle must have been evaluated. To this end points where the maximum number of iterations was reached or the probability of failure decreased with ascending water level were removed to obtain a monotonically increasing fragility curve.

**E: FORM algorithm**

The first order reliability method (FORM) is a method to iteratively calculate the probability of a limit state function ($Z(\mathbf{X}) \leq 0$) being exceeded given a set of independent random variables ($\mathbf{X}$) (Hasofer and Lind, 1974). The starting point for the iteration is arbitrary, but usually the mean of the variables is taken as the first point to evaluate ($\mathbf{x}^*$). The problem is first simplified by converting the random variables before each iteration into realisations of equivalent normally distributed variables ($\mathbf{x}'$) with an equivalent normal transformation (Rackwitz and Flessler, 1978).

$$\mu'_{X_i} = x_i^* - \sigma'_{X_i} * \Phi^{-1}[F(x_i^*)] \tag{E1}$$

$$\sigma'_{X_i} = \frac{\varphi\{\Phi^{-1}[F(x_i^*)]\}}{f(x_i^*)} \tag{E2}$$

Where $\mu'_{X_i}$ and $\sigma'_{X_i}$ are the mean and standard deviation of the equivalent normal distribution of variable $x_i$ in the point $\mathbf{x}^*$. Also f and F are the probability density function (PDF) and cumulative distribution function (CDF) of variable $x_i$ while $\varphi$ and $\Phi$ are the standard normal PDF and CDF.

The mean and standard deviation of the limit state function are evaluated by:

$$\mu_Z = Z(\mathbf{x}^*) + \sum_{i=1}^{n} \frac{\partial Z}{\partial X_i}(\mu'_{X_i} - x_i^*) \tag{E3}$$

$$\sigma_Z = \sqrt{\sum_{i=1}^{n} \left(\frac{\partial Z}{\partial X_i}\right)^2 {\sigma'_{X_i}}^2} \tag{E4}$$

With the mean and standard deviation calculated from the design point ($\mathbf{x}^*$) the reliability index ($\beta$) and influence factor of each variable ($\alpha_{X_i}$) are calculated.





$$\beta = \frac{\mu_Z}{\sigma_Z} \tag{E5}$$

$$\alpha_{X_i} = \frac{\partial Z}{\partial X_i} * \frac{\sigma'_{X_i}}{\sigma_Z} \tag{E6}$$

The point is updated by adjusting each variable based on the overall safety level ($\beta$) and the sensitivity of the limit state to the variable ($\alpha_{X_i}$):

$$x_i^* = \mu'_{X_i} - \alpha_{X_i}\beta\sigma'_{X_i} \tag{E7}$$

5   The process is repeated until the reliability index has converged and no longer changes significantly after an iteration.

While the method is effective there are limitations. It is not guaranteed FORM finds the design point with the highest probability of occurring but rather converges to a local minimum. Furthermore for FORM to converge the limit state function should be smooth without jumps or discontinuities. This complicated the implementation of for example macro stability as
10   when a different slip circle becomes critical there can be a sudden jump in the evaluation of the limit state function.



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


**Figures**

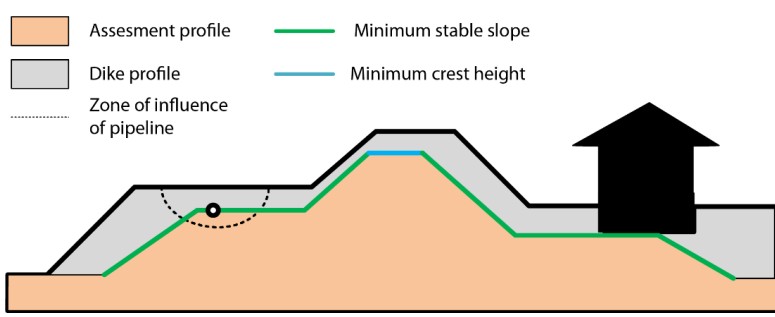

**Fig. 1 Assessment profile for a dike with NWOs (pipeline and house with basement). Adapted from figure A.4 of the current Dutch guidelines (Rijkswaterstaat, 2016)**

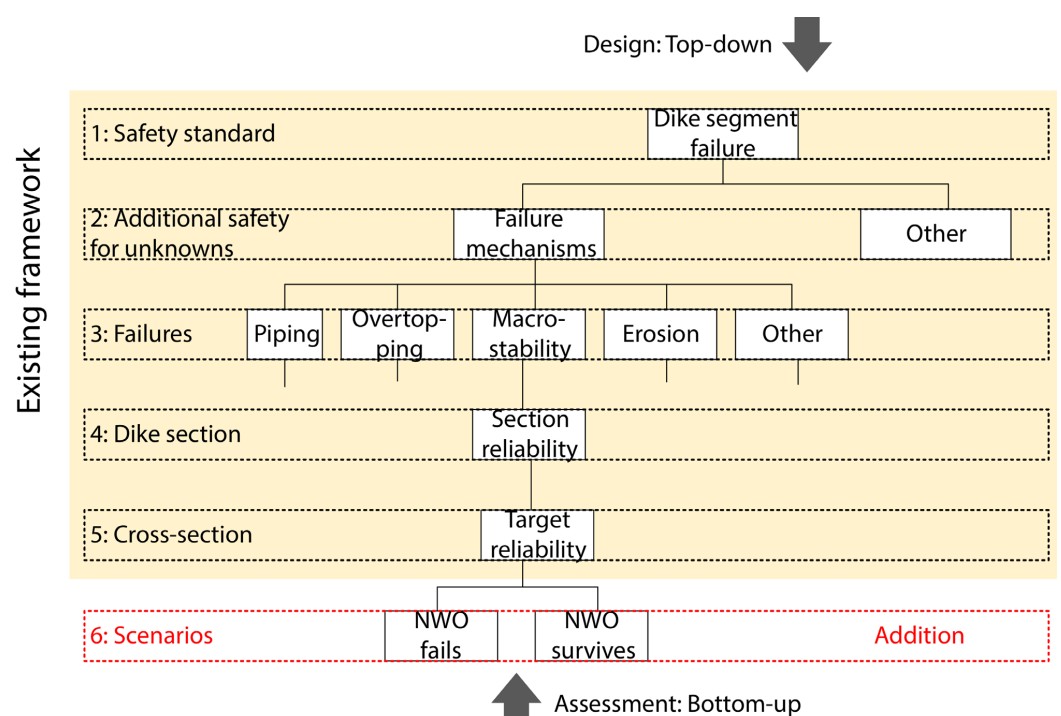

**Fig. 2 A framework for a detailed assessment and design of a dike with multifunctional elements. The yellow section is the existing framework while the last step in *red* denotes the addition of scenarios (e.g. a failed NWO and functioning NWO) to conform to a risk-based approach.**

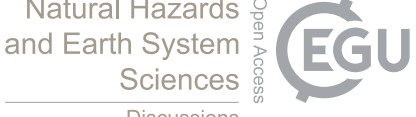


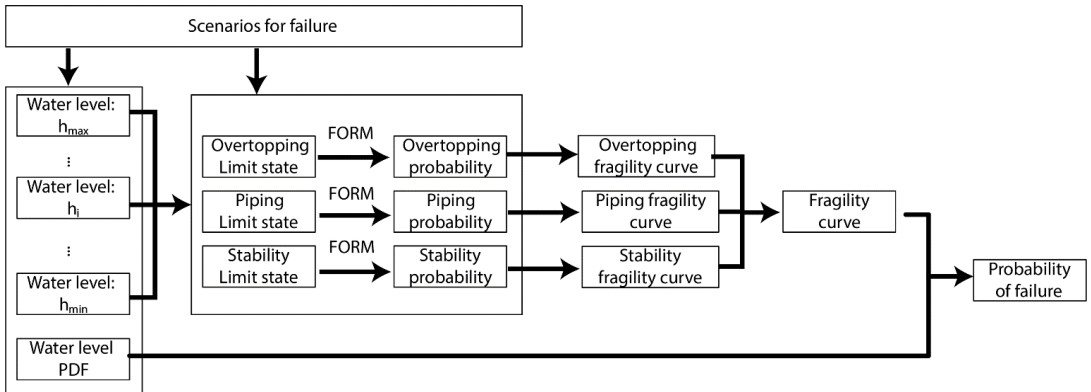

**Fig. 3 The probabilistic procedure for calculation the probability of failure of a dike cross-section in this study**

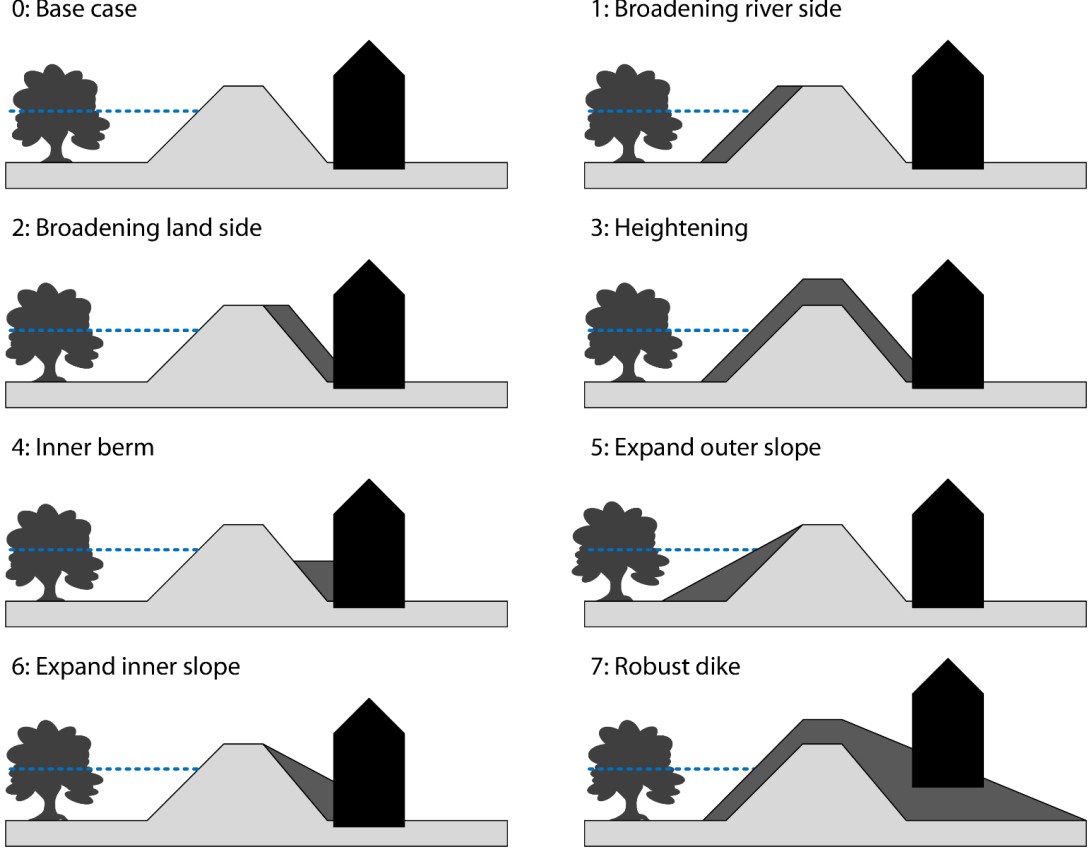

**Fig. 4 Case studies for comparing the conservative and the new probabilistic approach in this study**

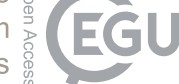
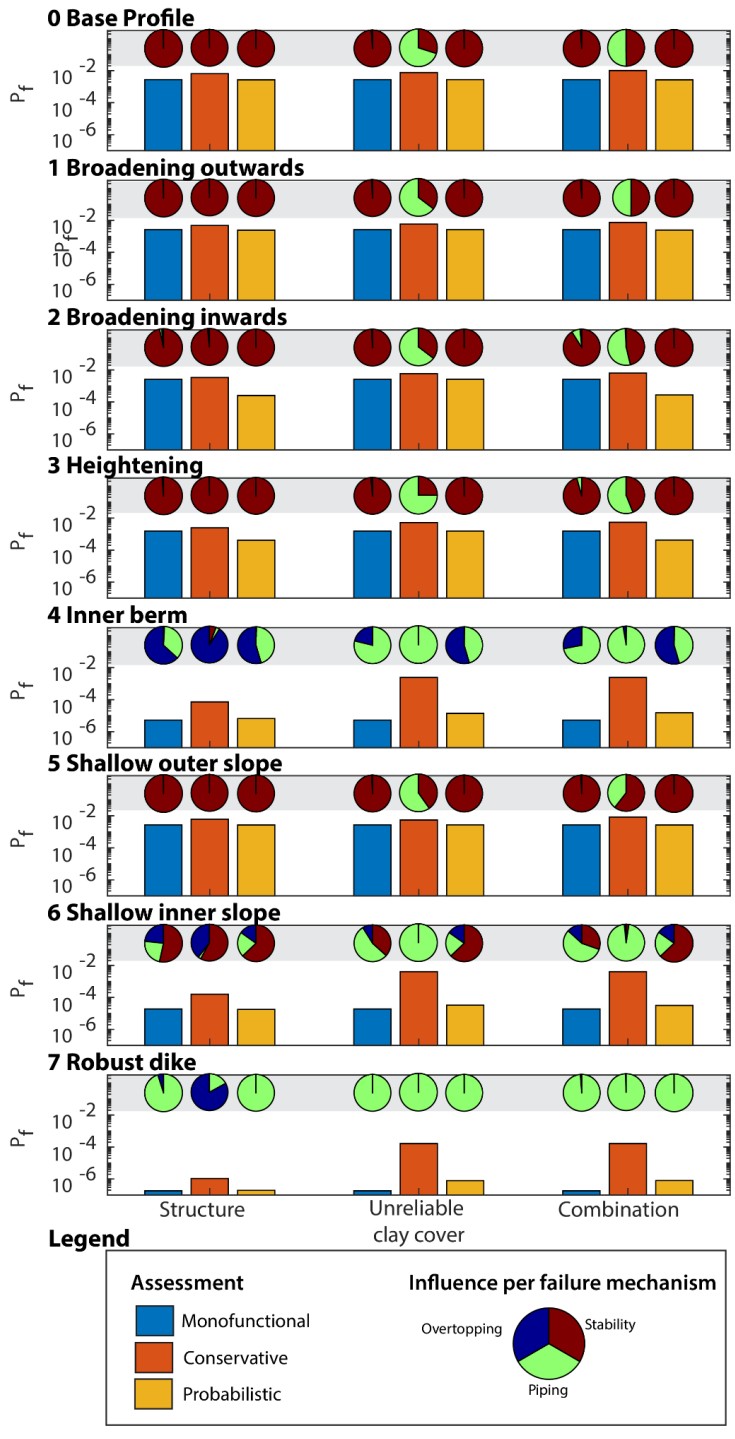

**Fig. 5 The probability of failure (P$_f$) for every dike profile (0 to 7) assessed as a monofunctional dike (blue bar), a multifunctional dike with a conservative approach (orange bar) and a multifunctional dike using a probabilistic approach (yellow bar) in the situation where a structure is present (left), an impaired clay cover on the flood plain could be present (middle) and both a structure and unreliable clay cover are present (right). The influence of the three failure mechanisms overtopping (blue), piping (green) and stability (red) is given per bar with a pie chart.**





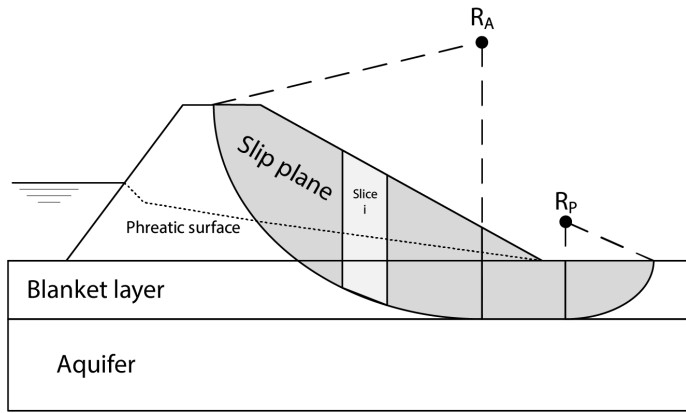

**Fig. D1 Schematisation of the slip plane and phreatic surface used for the macro stability calculation**



Tables

**Table 1 The different approaches for assessing the cross-section of a multifunctional dike in this study**

| Approach | | Assumptions | Example |
|---|---|---|---|
| Mono-functional | | • No functions present | |
| Multi-functional | Conservative | • Functions are always in the critical state for a given failure mechanisms<br>• Dike zones affected by the functions are omitted from the profile | |
| | Probabilistic | • Uncertainty of functions split into scenarios (e.g. present or absent)<br>• Each scenario has a probability of occurring | Scenario 1, Probability= X%<br><br>Scenario 2, Probability=100-X% |

**Table 2 Overview of failure mechanisms and corresponding methods**

| Failure mechanism | Description | Limit state function | Method |
|---|---|---|---|
| Overflow and overtopping | Excessive flow of water over the dike with severe inundation of the hinterland as a result, possibly by erosion of the revetment and soil on the crest and inner slope leading to a dike breach | $q_c - q$ | Overtopping: van der Meer et al. (2016), TAW (2002), Waal (1999) |
| Piping | Erosion of soil particles under the dike as a result of seepage. This in turn leads to collapse of the dike and failure by inundation of the hinterland. | $H_c - H$ | Ground water: TAW (2004)<br>Erosion: Sellmeijer et al. (2011) |
| Macro stability | Loss of slope stability as the dike becomes saturated. The collapse of the dike results in inundation of the hinterland | $\Sigma M_R - \Sigma M_S$ | Ground water: TAW (2004)<br>Slope stability: Van (2001) |

**Table 3 Variation in parameters between reinforcement strategies**

| Profile nr. | Inner slope [-] | Outer slope [-] | Crest height [m+REF] | Berm width [m] | Crest width [m] | Flood plain length [m] | Max. overtopping rate (μ, σ)* [l/m/s] | | | | | |
|---|---|---|---|---|---|---|---|---|---|---|---|---|
| | | | | | | | House intact | | House collapsed | | No house | |
| 0 | 1:2.5 | 1:3 | 5.5 | 0 | 5 | 100 | - | - | - | - | 100 | 120 |
| 1 | 1:2.5 | 1:3 | 5.5 | 0 | 10 | 95 | - | - | - | - | 100 | 120 |
| 2 | 1:2.5 | 1:3 | 5.5 | 0 | 10 | 100 | 70 | 80 | 0.1 | 0 | 100 | 120 |
| 3 | 1:2.5 | 1:3 | 6.5 | 0 | 5 | 97 | 70 | 80 | 0.1 | 0 | 100 | 120 |
| 4 | 1:2.5 | 1:3 | 5.5 | 15 | 5 | 100 | 70 | 80 | 0.1 | 0 | 100 | 120 |
| 5 | 1:2.5 | 1:4.5 | 5.5 | 0 | 5 | 91.75 | - | - | - | - | 100 | 120 |
| 6 | 1:4 | 1:3 | 5.5 | 0 | 5 | 100 | 70 | 80 | 0.1 | 0 | 100 | 120 |
| 7 | 1:10 | 1:3 | 6.5 | 0 | 5 | 97 | 70 | 80 | 0.1 | 0 | 100 | 120 |

*parameters of the lognormal distribution based on (van Hoven, 2015)




**Table A1 The standard geometry parameters for the dikes in the hypothetical case-study**

| Symbol | Description | Distribution | Parameters | |
|---|---|---|---|---|
| | | | μ | σ |
| $z_{hinter}$ | elevation of the hinterland [m] above REF | Deterministic | 0 | - |
| $z_{crest}$ | elevation of the crest [m] above REF | Deterministic | 5.5 | - |
| $z_{fore}$ | elevation of the foreshore (at the dike toe) [m] above REF | Deterministic | 0 | - |
| $z_{deep}$ | the average bed level, [m] above NAP along the fetch of the wind | Deterministic | -0.8 | - |
| $\tan(\alpha_{in})$ | inner slope angle [-] | Deterministic | 1/2.5 | - |
| $\tan(\alpha_{out})$ | outer slope angle [-] | Deterministic | 1/3 | - |
| $B_{crest}$ | crest width [m] | Deterministic | 5 | - |
| $L_f$ | length of the foreshore | Lognormal | 100 | 10 |

**Table A2 Standard parameters of the blanket layer for the dikes in the hypothetical case-study**

| Symbol | Description | Distribution | Parameters | |
|---|---|---|---|---|
| | | | μ | σ |
| $d_{blanket}$ | blanket layer thickness [m] | Lognormal | 2 | 0.6 |
| $\gamma_{sat,blanket}$ | saturated volumetric weight of the blanket layer [kN/m³] | Normal | 18.8 | 0.1 |
| $k_{blanket}$ | specific conductivity of the blanket layer [m/s] | Lognormal | 2.00E-08 | 2.00E-08 |
| $ch_{blanket}$ | cohesion of blanket material [kN/m²] | Deterministic | 0 | 0 |
| $\phi_{blanket}$ | Friction angle of blanket material [deg] | Normal | 28 | 4.5 |

**Table A3 Standard parameters of the aquifer layer for the dikes in the hypothetical case-study**

| Symbol | Description | Distribution | Parameters | |
|---|---|---|---|---|
| | | | μ | σ |
| $d_{aquifer}$ | Aquifer layer thickness [m] | Deterministic | 30 | |
| $\gamma_{sat,aquifer}$ | saturated volumetric weight of the aquifer layer [kN/m³] | Normal | 18 | 0.1 |
| $\eta$ | drag factor/White's coefficient [-] | Deterministic | 0.25 | |
| $\theta$ | bedding angle [rad] | Deterministic | 0.61 | |
| $d_{70}$ | 70%-percentile of the grain size distribution [m] | Lognormal | 3.07E-04 | 4.61E-05 |
| $k_{aquifer}$ | specific conductivity aquifer [m/s] | Lognormal | 4.86E-04 | 2.82E-04 |
| $ch_{aquifer}$ | cohesion of aquifer material [kN/m²] | Deterministic | 0 | 0 |
| $\phi_{aquifer}$ | Friction angle of aquifer material [deg] | Deterministic | 31.3 | 4.5 |

**Table A4 Standard parameters for the dike soil material for the dikes in the hypothetical case-study**

| Symbol | Description | Distribution | Parameters | |
|---|---|---|---|---|
| | | | μ | σ |
| $\gamma_{sat,core}$ | saturated volumetric weight of the dike core [kN/m³] | Normal | 18.2 | 0.1 |
| $\gamma_{dry,core}$ | dry volumetric weight of the core[kN/m³] | Normal | 13.1 | 0.1 |
| $ch_{core}$ | cohesion of core material [kN/m²] | Deterministic | 0 | 0 |
| $\phi_{core}$ | Friction angle of core material [deg] | Normal | 33 | 4.5 |



**Table A5 Standard hydraulic load and resistance parameters for the dikes in the hypothetical case-study**

| Symbol | Description | Distribution | Parameters | | Source |
|--------|-------------|--------------|------------|-----|--------|
| | | | μ | σ | |
| $\rho_w$ | density of water [kg/m³] | Normal | 1000 | 1 | Known constant |
| h | water level [m] above REF | Generalized extreme value | -2.5 | σ =1.5, ξ= -0.17 | Assumed |
| $\gamma_{break}$ | breaker index of waves [-] | Normal | 0.425 | 0.075 | Estimated (EurOtop, 2016;TAW, 2002) |
| $\gamma_f$ | roughness factor for an outer slope with grass [-] | Deterministic | 1 | - | (EurOtop, 2016;TAW, 2002) |
| $u_v$ | hourly wind speed at 10 m above the surface [m/s] | Gumbel | 16.8 | 1.6 | Assumed |
| $F_{max}$ | fetch [m] | Deterministic | 1800 | | Assumed |
| $q_c$ | critical overtopping discharge [l/m/s] | | | | (van Hoven, 2015) |
| | - No house (closed grass cover) | Lognormal | 100 | 120 | |
| | - Intact house (open grass cover) | Lognormal | 70 | 80 | |
| | - Collapsed house (no major overtopping allowed) | Lognormal | 0.1 | - | |

**Table B1  Description and values of variables in the overtopping and overflow limit state function**

| Variable | Description | Note |
|----------|-------------|------|
| $\alpha_{out}$ | Outer slope angle [-] | - |
| $\gamma_f$ | Friction factor for the outer slope [-] | 1 (TAW, 2002) |
| $H_s$ | Significant wave height [m] | *See Eq. (B6)* |
| $\xi_0$ | Iribaren number [-] | $\xi_0 = \dfrac{\tan(\alpha_{out})}{\sqrt{\dfrac{2\pi H_s}{gT_s^2}}}$ |
| $c_1$ | Factor for overtopping [-] | Normally distributed with μ=4.75 and σ=0.5 (TAW, 2002) |
| $c_2$ | Factor for overtopping [-] | Normally distributed with μ=-0.92 and σ=0.24 (TAW, 2002) |
| $m_{Bret,H}$ | Model factor for Bretschneider equation | Lognormally distributed with μ=1 and σ=0.27  (Diermanse, 2016) |
| $m_{Bret,T}$ | Model factor for Bretschneider equation | Lognormally distributed with μ=1 and σ=0.13 (Diermanse, 2016) |

5   **Table C1 Description and values of variables in the piping limit state function**

| Variable | Description | Destribution | Parameters | Unit |
|----------|-------------|--------------|------------|------|
| $\gamma_p$ | specific weight of sand particles | Deterministic | 26 | $\dfrac{kN}{m^3}$ |
| $\gamma_w$ | specific weight of water | Deterministic | 10 | $\dfrac{kN}{m^3}$ |
| $\eta$ | drag factor | Deterministic | 0.25 | - |
| $\theta$ | bedding angle [°] | Deterministic | 35 | - |
| $\dfrac{RD}{RD_m}$ | Relative density of the material compared to small-scale piping experiments | Determinsistic | 1 | - |
| $d_{70m}$ | Reference $d_{70}$ of the material used in small-scale piping experiments | Determinsistic | $2 * 10^{-4}$ | m |
| $m_p$ | Model factor for piping | Lognormal | $\mu = 1, \sigma = 0.12$ | - |