# Peer review of "Re-evaluating safety risks of multifunctional dikes with a probabilistic risk framework"

_Natural Hazards and Earth System Sciences, 2018_

## Referee Comment (RC1) · Anonymous Referee #1 · 17 Dec 2018

Aspects for the full review: 1. Does the paper address relevant scientific and/or technical questions within the scope of NHESS? (yes) 2. Does the paper present new data and/or novel concepts, ideas, tools, methods or results? (yes) 3. Are these up to international standards? (yes) 4. Are the scientific methods and assumptions valid and outlined clearly? (Generally yes, however only if the basic concept of probability based design of flood protection is well known to the reader...) 5. Are the results sufficient to support the interpretations and the conclusions? (yes) 6. Does the author reach substantial conclusions? (Yes, when it comes to defining the knowledge gap for further research and development.) 7. Is the description of the data used, the methods used, the experiments and calculations made, and the results obtained sufficiently complete and accurate to allow their reproduction by fellow scientists (traceability of results)? (In

the paper it should be explained more clearly what was exactly calculated and from which calculation followed which result. In general, however, this part is OK.) 8. Does the title clearly and unambiguously reflect the contents of the paper? (Yes) 9. Does the abstract provide a concise, complete and unambiguous summary of the work done and the results obtained? (yes) 10. Are the title and the abstract pertinent, and easy to understand to a wide and diversified audience? (yes) 11. Are mathematical formulae, symbols, abbreviations and units correctly defined and used? If the formulae, symbols or abbreviations are numerous, are there tables or appendixes listing them? (n/a) 12. Is the size, quality and readability of each figure adequate to the type and quantity of data presented? (See below) 13. Does the author give proper credit to previous and/or related work, and does he/she indicate clearly his/her own contribution? (yes) 14. Are the number and quality of the references appropriate? (yes) 15. Are the references accessible by fellow scientists? (In general yes, without checking every single reference) 16. Is the overall presentation well structured, clear and easy to understand by a wide and general audience? (Generally yes, however, a wide and general audience would need more detail on the probabilistic design concept and its implications when it comes to multifunctional dikes) 17. Is the length of the paper adequate, too long or too short? (adequate) 18. Is there any part of the paper (title, abstract, main text, formulae, symbols, figures and their captions, tables, list of references, appendixes) that needs to be clarified, reduced, added, combined, or eliminated? (See below) 19. Is the technical language precise and understandable by fellow scientists? (yes) 20. Is the English language of good quality, fluent, simple and easy to read and understand by a wide and diversified audience? (yes) 21. Is the amount and quality of supplementary material (if any) appropriate? (yes)

General comments: The paper is extremely relevant in the context of the design of flood protection under consideration of multifunctionality. In general it is well structured and contains all necessary information to follow the discussion. The paper could be improved by explaining more clearly what was actually calculated to allow a better understanding of results and findings. The word function is used regarding multifunctionality as well as mathematical function. Because this word is used very often in the text, the authors should revise the text if the context for the word function is always clear.

Specific comments: Page 1: L11ff: (While a traditional….) please define more precisely. As for now it does not become clear what the difference really is. Page 3: L18: "…to exclude flood defences with an insignificantly low failure probability… " If The probability to fail is insignificantly low, this would be a positive result. Why should such flood defences be excluded? Page 5: L11ff: The referral to table 1 gives the impression that either the set of analysed MFFDs or the respective calculations can be found in table 1. Neither is correct. Table 1 only shows (very generally) the differences of the approaches. Table 1: Please rethink: If the probability of occurrence in scenario 1 (additional function present) is x%, is then the probability for scenario 2 (additional function absent) really 100-x %? This seems to be a mistake. Otherwise this needs explanation in the text. Page 7: L28: is there really a hole presenting the profile or is there an empty space, the outer shape of the area of the additional function or something like this? L31f: Why is the probability of the absent structure chosen to be 1%? And why is this a conservative approach? Page 8: L3: …when the structure remains just outside of the profile (0,1,2)… Please explain: why does this not also apply to profiles 3 and 5?

Technical corrections: Page 1: L22: …, a better understanding… L26: "This is true…" please reformulate. Page 3: L4ff: please do not use the personal pronoun "we". Page 5: L12: …and the traditional… Table 1: …a given failure mechanism …probability of occurrence Page 7: L10: …by weighing… L25: please reformulate… L27: …2 two… ?; …present in which CASE the load… Page 8: L1: …berm [] both… L6: for better readability: …along the full length, the inclusion of uncertainty… L7f: reformulate: TRUE L34: reformulate: "risk of functions" Page 9: L1: personal pronoun "we"… see above L2f: Please revise the sentence for better understanding. Figure 3: Please reformulate the caption: …"for calculation the probability"…

---

## Referee Comment (RC2) · Anonymous Referee #2 · 6 Jan 2019

The manuscript by Marijnissen et al. applies the probabilistic dike failure assessment framework to the multi-functional dikes. These dikes incorporate besides the genuine flood protection function other functional elements such as for instance natural vegetation and build structures (houses). The authors compare the probabilistic assessment of multi-functional dikes considering the failure probability of dikes due to overtopping, piping and macro-instability and also the probability of failure of additional functions to the mono-functional assessment and the conservative assessment. The latter considers the failure of additional functions (with the probability of 1) and represents the current engineering practice of failure probability assessment of multi-functional dikes. The analysis of various dike reinforcement scenarios in the presence of either a build-structure or vegetation or both suggests that the probabilistic assessment of

functions failure results in overall lower failure probability compared to the conservative approach. Compared to the mono-functional assessment the failure probability may be higher or lower for a multi-functional dike depending on the net positive or negative effect of the function and the interplay with the reinforcement measure. The manuscript is well-structured, presents a novel advancement of the methodology and reaches substantial conclusions. It is mostly well-written, though some sections need to be made clearer (see comments below). However, I believe the authors can further strongly improve the manuscript with regards to two aspects.

(1) The section P7-L18-32 describes the effect of two additional functions (vegetation and build-structure) on dike stability. I found this description rather cryptic and unclear. It should be significantly improved. It is not clear, how either of the functions affects each breach mechanism. Does the build structure affects only macro-instability due to additional weight? Is there effect on piping, e.g. due to longer pipe length needed to induce a dike failure? What do you mean by the insignificant amount of overtopping $q<0.1$ is acceptable? What does this have to do with the structure or absence of a structure? How one should imagine a scenario (with the annual probability of 1%), where a house disappears creating a hole at its location with the dike being intact (!). This is not clear to me.

(2) The second aspect is related to the first one and concerns the results of failure probability calculations (Fig. 5) and in particular the influence of different breach mechanisms. As the role of functions for various breach mechanisms was not clarified in details, it is very difficult to understand the effect of considering these functions on the distribution and changes of breach mechanisms presented in pie charts in Fig. 5. Unfortunately, the authors only scratch the surface and leave much of the presented results undiscussed and not analysed in-depth. I would appreciate a much more detailed analysis and discussion of the effects of (a) function failure and (b) reinforcement scenarios onto the role of breach mechanisms.

Finally, the authors mention in the text that uncertainties where somehow considered

by considering scenario 7 "Robust dike". I did not understand this and do not see that uncertainties (of whatever nature) are considered here. Actually, the study is self-contained and there is no need to assess uncertainties as the probabilistic analysis already incorporates the uncertainties of various model parameters.

Abstract is poorly written and is not self-explaining. L11-14 are unclear for someone who has not read the paper and comes with general, though profound knowledge on flood risk.

In overall, I rate this study as very solid and believe that after addressing the two major issues and a few minor comments below it can make an interesting and significant contribution to the research on probabilistic assessment of dike failures and flood risk assessment.

Minor issues: Introduction: The text is somewhat doggerel and needs a careful revision. (e,g, P1-L26-35 and comments below)

P1-L19: risk of floods is not increasing everywhere. One should differentiate. "these cataastrophes" – you are talking of risk in general and not about some specific catastrophes.

P1-L21: "Risk based approaches have been" used not "performed". L22: remove "the" before "understanding".

P1-L36-37: revise the sentence.

P2-L1: Is this really true? The nation-wide risk assessment for England and Wales (Hall et al., 2003, 2005) also used probabilistic approach to assessment of protection level/failure probability.

P2-L35: Reference Hinkel et al. is missing in the reference list.

P3-L1: what is a 'cohesive' framework?

P3-L40 – P4-L1: as you mention, a conservative approach is usually taken assuming

the NWO to be in the most critical state. Make clear that the actual probability of failure of the NWO is thus not considered.

P4-L9ff: in general, it seems that the vast majority of literature sources used in the manuscript is of Dutch origin. Nevertheless, there is also some relevant literature outside. E.g. the use of limit state functions and fault trees for flood defence assessment and hazard/risk assessment was performed by Kortenhaus (2003), Apel et al. (2004), Dawson & Hall (2006), Vorogushyn et al. (2009, 2010).

P4-L28: what is WBI2017?

P5-L19: also Vorogushyn et al. (2009) compiled the statistics on dike failures from a few previous studies

Eq.3.6: Use h=0 as the lower limit of the integral. –Inf dies not make sense for water levels.

P8-L3: Is this correct that the effect of the function is limited in the scenario 2? The yellow bar is significantly lower than for the monofunctional assessment! At P7-L40 you mentioned that in the scenario 2 there is a significant positive effect of the structure. Please, check.

P9-L29-30: The sentence and the message is unclear to me. The list of references is not carefully formatted. Temmermann et al., journal missing. Move the quation for the Iribaren number from Table B1 into the B-section prior or after Eq. B10.

References:

Apel, H., Thieken, A. H., Merz, B., and Blöschl, G.: Flood risk assessment and associated uncertainty, Nat. Hazards Earth Syst. Sci., 4, 295–308, 2004.

Kortenhaus, A. (2003): Probabilistische Methoden für Nordseedeiche. Dissertation. TU Braunschweig, Germany.

Dawson, R. J. and Hall, J. W.: Adaptive importance sampling forrisk analysis of complex infrastructure systems, Proc. R. Soc. A.,462, 3213–3499, doi:10.1098/rspa.2006.1720, 2006.

Vorogushyn, S., Merz, B., Apel, H., 2009. Development of dike fragility curves for piping and micro-instability breach mechanisms. Natural Hazards and Earth System Sciences 9, 1383–1401.

Vorogushyn, S., Merz, B., Lindenschmidt, K.-E., Apel, H., 2010. A new methodology for flood hazard assessment considering dike breaches. Water Resources Research 46, W08541. http://dx.doi.org/10.1029/2009WR008475.

---

## Author Comment (AC1) · 11 Feb 2019

We thank reviewer 1 for the constructive and helpful comments and suggestions. We hope he/she will be satisfied with the coming changes of the manuscript.

General comments:

Comment 1: The paper is extremely relevant in the context of the design of flood protection under consideration of multifunctionality. In general it is well structured and contains all necessary information to follow the discussion

Response: We thank the reviewer for this nice comment.

Comment 2: The paper could be improved by explaining more clearly what was actually

calculated to allow a better understanding of results and findings.

Response: We agree that further explanations on the methods and findings are needed. The section describing how mutlifunctionality is implemented in the case-study is (being) revised entirely as well as further explanations on the reinforcement strategies. Furthermore additional explanations on the results will be provided in the revised manuscript. Finally the abstract is being revised entirely based on suggestions from reviewer 2 as well to make it clearer what was calculated.

Comments 3: The word function is used regarding multifunctionality as well as mathematical function. Because this word is used very often in the text, the authors should revise the text if the context for the word function is always clear.

Response: This is a good point. The word "function" in the context of multifunctionality will be changed to "multifunctional use" or "multifunctional elements". Function in the context of mathematical functions are usually part of a larger definition like Probability Density Function (PDF) which require no further context.

Specific comments:

Comment 4: Page 1: L11ff: (While a traditional ...) please define more precisely. As for now it does not become clear what the difference really is.

Response: This was indeed not clear from the abstract. Based on the suggestion of reviewer 2 as well the abstract is being revised to be more self-explanatory.

Comment 5: Page 3: L18: "... to exclude flood defences with an insignificantly low failure probability ... " If the probability to fail is insignificantly low, this would be a positive result. Why should such flood defences be excluded?

Response: The goal of an assessment is to check whether it meets the safety standards. If the dike passes the basic assessment in can be excluded from further detailed and tailored assessments for that failure mechanism as it is already considered to be safe. To reflect this the word "exclude" can be replaced by "approve".

Comment 6: Page 5: L11ff: The referral to table 1 gives the impression that either the set of analysed MFFDs or the respective calculations can be found in table 1. Neither is correct. Table 1 only shows (very generally) the differences of the approaches.

Response: Table 1 is meant to introduce the reader to the different approaches as you correctly identified. The sentence will be reformulated to: "a set of MFFDs is assessed with the new probabilistic approach and the traditional conservative approach (see Table 1 for the approaches)."

Comment 7: Table 1: Please rethink: If the probability of occurrence in scenario 1 (additional function present) is x%, is then the probability for scenario 2 (additional function absent) really 100-x %? This seems to be a mistake. Otherwise this needs explanation in the text.

Response: x will be changed into P (from probability) in scenario 1 and 1-P in scenario 2. Since only 2 scenarios are considered the probability of scenario 1 + the probability of scenario 2 must equal 1 or 100%. The % sign may have led to confusion and will therefore be removed.

Comment 8: Page 7: L28: is there really a hole presenting the profile or is there an empty space, the outer shape of the area of the additional function or something like this?

Response: In principle all space occupied by the additional function becomes empty. This was referred to as a hole because the outer edge is no longer grass, but rather loose soil. In the revised version of this section the word hole will not be used as it may convey a different message. This section will be revised entirely (also based on a comment by reviewer 2).

Comment 9: L31f: Why is the probability of the absent structure chosen to be 1%? And why is this a conservative approach?

Response: This number has been subject to some discussion. Initially the reliability

requirement for housing structures in the Netherlands was taken (P_absent = 1E-5), but that would only reflect the structural reliability. Another approach was to look at the designed lifespan of houses which is 50 years (P_absent = 0.02) but this neglects the fact many structures are renovated rather than destroyed. According to van der Flier and Thomsen (2006) 0.13 and 0.23% of houses are demolished annually in the Netherlands. Based on this 1% was chosen as an conservative order of magnitude estimation of the house being demolished during a high water event. This explanation will be added in the revised manuscript.

Comment 10: Page 8: L3: ... when the structure remains just outside of the profile (0,1,2)... Please explain: why does this not also apply to profiles 3 and 5?

Response: Thank you for pointing this out. This sentence should have referred to 0,1 and 5 not 0,1 and 2. It does not apply to 2 and 3 because here soil is replaced by additional weight of the structure leading to a net positive effect on stability. Because the explanation of the findings were confusing to the reader, this section has been rewritten for the revised manuscript.

Technical corrections

Comment 11: Page 1: L22: ..., a better understanding... L26: "This is true..."please reformulate. Page 3: L4ff: please do not use the personal pronoun "we". Page 5: L12: ...and the traditional... Table 1: ...a given failure mechanism ... probability of occurrence Page 7: L10: ... by weighing... L25: please reformulate... L27: ...2 two...?; ...present in which CASE the load... Page 8: L1: ... berm [] both... L6: for better readability: ...along the full length, the inclusion of uncertainty... L7f: reformulate: TRUE L34: reformulate: "risk of functions" Page 9: L1: personal pronoun "we"... see above L2f: Please revise the sentence for better understanding. Figure 3: Please reformulate the caption: ...for calculation the probability"...

Response: We thank the reviewer for pointing out these corrections. These will be implemented in the revision.

References:

van der Flier, C., and Thomsen, A.: Life cycle of dwellings: Analysis and assessment of demolition by Dutch housing associations, International Conference ENHR, Ljubljana, Slovenia, 2-5 July 2006; Workshop 7, Physical Aspects of Design and Regeneration, 2006,

———————————————

---

## Author Comment (AC2) · 11 Feb 2019

We would like to thank the reviewer for his/her thorough feedback on the paper. We are sure this feedback will be of great help during the revision and hope the reviewer will be satisfied with the changes.

General comments:

Comment 1: ... The manuscript is well-structured, presents a novel advancement of the methodology and reaches substantial conclusions. It is mostly well-written, though some sections need to be made clearer (see comments below). However, I believe the authors can further strongly improve the manuscript with regards to two aspects.

Response: We thank the reviewer for this comment. Indeed we think that by following

the suggestions the paper will improve.

Comment 2: (1) The section P7-L18-32 describes the effect of two additional functions (vegetation and build-structure) on dike stability. I found this description rather cryptic and unclear. It should be significantly improved. It is not clear, how either of the functions affects each breach mechanism. [1] Does the build structure affects only macro-instability due to additional weight? [2] Is there effect on piping, e.g. due to longer pipe length needed to induce a dike failure? [3] What do you mean by the insignificant amount of overtopping q<0.1 is acceptable? [4] What does this have to do with the structure or absence of a structure? [5] How one should imagine a scenario (with the annual probability of 1%), where a house disappears creating a hole at its location with the dike being intact (!). This is not clear to me.

Response: We agree that this section can be made clearer and plan to revise it based on this suggestion to make it clearer. The individual points are addressed here:

[1] The built structure is schematised by 2 effects in the model: 1) its additional weight and 2) a decrease in the maximum allowable overtopping rate.

[2] No, in our cases the structure does not extend into the aquifer and thus has little effect on piping. Furthermore one could argue the presence of a structure inhibits the well from emerging at the structure's location but a pipe could emerge just beside the structure instead along its outer wall.

[3] & [4] During overtopping the outer layer on the landward side of the dike should not erode during design conditions. When a structure is present it is effectively a discontinuity in the outer cover where water can more easily erode soil. This is reflected in a lower critical overtopping rate (q). When the structure is absent (see Table 1) there is presumed to be no grass cover but instead loose bare soil that will almost immediately start eroding. Since it is statistically impossible to rule out any overtopping (q=0 l/m/s) it is practise in the Netherlands to use q=0.1 l/m/s as a threshold value if effectively no erosion resistance can be expected and thus practically no overtopping is allowed

during design conditions.

[5] The situation considered is not necessarily a collapse of the structure, but can also be when the structure is removed temporarily (e.g. a renovation or demolished intentionally to construct another). See also our response to reviewer 1. We will make sure to clarify this distinction in the revision.

Comment 3: The second aspect is related to the first one and concerns the results of failure probability calculations (Fig. 5) and in particular the influence of different breach mechanisms. As the role of functions for various breach mechanisms was not clarified in details, it is very difficult to understand the effect of considering these functions on the distribution and changes of breach mechanisms presented in pie charts in Fig. 5. Unfortunately, the authors only scratch the surface and leave much of the presented results undiscussed and not analysed in-depth. I would appreciate a much more detailed analysis and discussion of the effects of (a) function failure and (b) reinforcement scenarios onto the role of breach mechanisms.

Response: We agree with reviewer 2 and we will explain this more in-depth by elaborating further on the reinforcements in section 3.4. Furthermore we plan to expand the results section to analyse the role of function failure for each of the reinforcement strategies as well as with additional figure(s).

Comment 4 Finally, the authors mention in the text that uncertainties where somehow considered by considering scenario 7 "Robust dike". I did not understand this and do not see that uncertainties (of whatever nature) are considered here. Actually, the study is self-contained and there is no need to assess uncertainties as the probabilistic analysis already incorporates the uncertainties of various model parameters.

Response: Indeed this sentence and its message were unclear. It was shown in the calculations that as the reliability of the dike increases, the influence of the uncertainty introduced by the functions has less influence on the failure probability. As the robust dike has the highest reliability, this effect was most clearly visible through this dike. The
sentence will be revised.

Comment 5: Abstract is poorly written and is not self-explaining. L11-14 are unclear for someone who has not read the paper and comes with general, though profound knowledge on flood risk.

Response: Based on your and reviewer 1's comment the abstract will be rewritten for the revision.

Comment 6: In overall, I rate this study as very solid and believe that after addressing the two major issues and a few minor comments below it can make an interesting and significant contribution to the research on probabilistic assessment of dike failures and flood risk assessment.

Response: We are happy with the comments and motivated to address the issues you have pointed out.

Minor issues

Comment 7: Introduction: The text is somewhat doggerel and needs a careful revision. (e,g, P1-L26-35 and comments below)

Response: The section between P1-L26-35 will be shortened to be more concise.

Comment 8: P1-L19: risk of floods is not increasing everywhere. One should differentiate. "these cataastrophes" – you are talking of risk in general and not about some specific catastrophes.

Response: While flood risks are in general increasing due to a combination of climate change (sea-level rise and extreme rainfall) and economic developments in deltas, I can imagine that in specific regions this is not the case. I will specify that only applies for a majority of flood prone regions. I will also replace the word catastrophes with floods for clarity.

Comment 9: P1-L21: "Risk based approaches have been" used not "performed". L22:

remove "the" before "understanding".

Response: Thank you for pointing out these corrections

Comment 10: P1-L36-37: revise the sentence.

Response: The sentence will be revised to: "Before the Dutch Water Act was revised the protection level of flood defences was defined only by the exceedance probability of extreme conditions which the flood defence is designed to withstand (Van der Most et al., 2014)"

Comment 11: P2-L1: Is this really true? The nation-wide risk assessment for England and Wales (Hall et al., 2003, 2005) also used probabilistic approach to assessment of protection level/failure probability.

Response: While a probabilistic approach was certainly used there, by our knowledge it was not legally required to do so. We will check this for the revision.

Comment 12: P2-L35: Reference Hinkel et al. is missing in the reference list.

Response: Hinkel et al. will be added to the reference list.

Comment 13: P3-L1: what is a 'cohesive' framework?

Response: The word should have been coherent, not cohesive. We will make the correction.

Comment 14: P3-L40 – P4-L1: as you mention, a conservative approach is usually taken assuming the NWO to be in the most critical state. Make clear that the actual probability of failure of the NWO is thus not considered.

Response: The word "actual" will been inserted to make this clear: "... because the actual probability of multifunctional elements being in a critical state is not considered".

Comment 15: P4-L9ff: in general, it seems that the vast majority of literature sources used in the manuscript is of Dutch origin. Nevertheless, there is also some relevant

literature outside. E.g. the use of limit state functions and fault trees for flood defence assessment and hazard/risk assessment was performed by Kortenhaus (2003), Apel et al. (2004), Dawson & Hall (2006), Vorogushyn et al. (2009, 2010).

Response: Indeed we could have used more international sources. The majority of references are of Dutch origin as the starting point of the research was the Dutch guidelines and sources/studies to support them. We will look to incorporate some suggested literature in the revision.

Comment 16: P4-L28: what is WBI2017?

Response: WBI2017 is the official abbreviation of the current Dutch assessment tools. References to WBI2017 should be changed to "the official Dutch assessment frame-work for flood defences"

Comment 17: P5-L19: also Vorogushyn et al. (2009) compiled the statistics on dike failures from a few previous studies

Response: Thank you again for the suggested literature. We will try to incorporate the suggestion in the manuscript.

Comment 18: Eq.3.6: Use h=0 as the lower limit of the integral. –Inf dies not make sense for water levels.

Response: If h would refer to water depth the lower limit would indeed make more sense to be 0. However since h can be negative in some reference systems (e.g. -1m +MSL which would be 1m below Mean Sea Level), h=-Inf is appropriate here.

Comment 19: P8-L3: Is this correct that the effect of the function is limited in the scenario 2? The yellow bar is significantly lower than for the monofunctional assessment! At P7-L40 you mentioned that in the scenario 2 there is a significant positive effect of the structure. Please, check.

Response: As also pointed out by reviewer 1, this sentence should have referred to

profiles 0,1 and 5 not 0,1 and 2. It does not apply to 2 and 3 because here soil is replaced by additional weight of the structure leading to a net positive effect on stability. We will correct this.

Comment 20: P9-L29-30: The sentence and the message is unclear to me.

Response: The message was that before the new Water Act probabilistic assessments could be used but there was no obligation to do so. Now a probabilistic assessment is required and naturally more probabilistic assessments are being used. The sentence will be revised.

Comment 21: The list of references is not carefully formatted. Temmermann et al., journal missing.

Response: The journal will be added. Thank you for spotting this.

Comment 22: Move the equation for the Iribaren number from Table B1 into the B-section prior or after Eq. B10.

Response: The equation will be moved after Eq. B7.
* * *

---

## Author Response (AR1)

**Response to reviewers' comments to the manuscript "Re-evaluating safety risks of multifunctional dikes with a probabilistic risk framework" by Richard Marijnissen et al.**

We thank reviewer 1 for his constructive and helpful comments and suggestions. We hope he/she is satisfied with the changes of the manuscript.

**Reviewer 1:**

*General comments:*

**Comment 1:**

> The paper is extremely relevant in the context of the design of flood protection under consideration of multifunctionality. In general it is well structured and contains all necessary information to follow the discussion

> **Response:**
> We thank the reviewer for this nice comment.

**Comment 2:**

> The paper could be improved by explaining more clearly what was actually calculated to allow a better understanding of results and findings.

> **Response:**
> We agree that further explanations on the methods and findings are needed. The section describing how multifunctionality is implemented in the case-study has been revised entirely with explanations on the reinforcement strategies. Furthermore additional analyses of the results are provided in the revised manuscript with an additional figure (Fig. 6). Finally the abstract has been revised entirely based on suggestions from reviewer 2 as well to make it clearer what was calculated. Please see the marked-up document for details of the changes.

**Comments 3:**

> The word function is used regarding multifunctionality as well as mathematical function. Because this word is used very often in the text, the authors should revise the text if the context for the word function is always clear.

> **Response:**
> This is a good point. The word "function" in the context of multifunctionality will be changed to "multifunctional use" or "multifunctional elements" where possible. Function in the context of mathematical functions are usually part of a larger definition like Probability Density Function (PDF) which require no further context. As can be seen in the marked-up manuscript many instances of the word "function" have been replaced.

*Specific comments:*

**Comment 4:**

> Page 1: L11ff: (While a traditional ...) please define more precisely. As for now it does not become clear what the difference really is.

**Response:**

This was indeed not clear from the abstract. Based on the suggestion of reviewer 2 as well the abstract has been revised entirely.

**Comment 5:**

Page 3: L18: "... to exclude flood defences with an insignificantly low failure probability ... " If the probability to fail is insignificantly low, this would be a positive result. Why should such flood defences be excluded?

**Response:**

The goal of an assessment is to check whether it meets the safety standards. If the dike passes the basic assessment in can be excluded from further detailed and tailored assessments for that failure mechanism as it is already considered to be safe. To reflect this the word "exclude" has been replaced by "approve".

**Comment 6:**

Page 5: L11ff: The referral to table 1 gives the impression that either the set of analysed MFFDs or the respective calculations can be found in table 1. Neither is correct. Table 1 only shows (very generally) the differences of the approaches.

**Response:**

Table 1 is meant to introduce the reader to the different approaches as you correctly identified. The sentence has been reformulated to: "a set of MFFDs is assessed with the new probabilistic approach and the traditional conservative approach (see Table 1 for the approaches)."

**Comment 7:**

Table 1: Please rethink: If the probability of occurrence in scenario 1 (additional function present) is x%, is then the probability for scenario 2 (additional function absent) really 100-x %? This seems to be a mistake. Otherwise this needs explanation in the text.

**Response:**

x has been changed into P (from probability) in scenario 1 and 1-P in scenario 2. Since only 2 scenarios are considered the probability of scenario 1 + the probability of scenario 2 must equal 1 or 100%. The % sign may have led to confusion and has therefore been removed.

**Comment 8:**

Page 7: L28: is there really a hole presenting the profile or is there an empty space, the outer shape of the area of the additional function or something like this?

**Response:**

In principle all space occupied by the additional function becomes empty. This was referred to as a hole because the outer edge is no longer grass, but rather loose soil. In the revised version the word hole has not been used as it may convey a different message. This section has been revised entirely (also based on a comment by reviewer 2).

**Comment 9:**

L31f: Why is the probability of the absent structure chosen to be 1%? And why is this a conservative approach?

**Response:**

This number has been subject to some discussion. Initially the reliability requirement for housing structures in the Netherlands was taken (P_absent = 1E-5), but that would only reflect the

structural reliability. Another approach was to look at the designed lifespan of houses which is 50 years (P_absent = 0.02) but this neglects the fact many structures are renovated rather than destroyed. According to van der Flier and Thomsen (2006) 0.13 and 0.23% of houses are demolished annually in the Netherlands. Based on this 1% was chosen as an conservative order of magnitude estimation of the house being demolished during a high water event. This explanation has been added in the revised manuscript as:

*"The probability the structure is absent during a high water event is estimated to be 1%. This probability is based on the percentage of houses demolished in the Netherlands annually which has varied between 0.13 and 0.23% per year (van der Flier and Thomsen, 2006) rather than the probability of structural failure of the house."*

**Comment 10:**

Page 8: L3: ... when the structure remains just outside of the profile (0,1,2)... Please explain: why does this not also apply to profiles 3 and 5?

**Response:**
Thank you for pointing this out. This sentence should have referred to 0,1 and 5 not 0,1 and 2. It does not apply to 2 and 3 because here soil is replaced by additional weight of the structure leading to a net positive effect on stability. Because the explanation of the findings were confusing to the reader, this section has been rewritten for the revised manuscript.

*Technical corrections*

**Comment 11:**

Page 1: L22: ..., a better understanding... L26: "This is true..."please reformulate. Page 3: L4ff: please do not use the personal pronoun "we". Page 5: L12: ...and the traditional... Table 1: ...a given failure mechanism ... probability of occurrence Page 7: L10: ... by weighing... L25: please reformulate... L27: ...2 two...?; ...present in which CASE the load... Page 8: L1: ... berm [] both... L6: for better readability: ...along the full length, the inclusion of uncertainty... L7f: reformulate: TRUE L34: reformulate: "risk of functions" Page 9: L1: personal pronoun "we"... see above L2f: Please revise the sentence for better understanding. Figure 3: Please reformulate the caption: ...for calculation the probability"...

**Response:**
We thank the reviewer for pointing out these corrections. These have been implemented in the revision.

**References:**

van der Flier, C., and Thomsen, A.: Life cycle of dwellings: Analysis and assessment of demolition by Dutch housing associations, International Conference ENHR, Ljubljana, Slovenia, 2-5 July 2006; Workshop 7, Physical Aspects of Design and Regeneration, 2006,

**Reviewer 2:**

We would like to thank the reviewer for his thorough feedback on the paper. This feedback was of great help during the revision and hope the reviewer is satisfied with the changes.

*General comments:*

**Comment 1:**

... The manuscript is well-structured, presents a novel advancement of the methodology and reaches substantial conclusions. It is mostly well-written, though some sections need to be made clearer (see comments below). However, I believe the authors can further strongly improve the manuscript with regards to two aspects.

**Response:**
We thank the reviewer for this comment. Indeed we think that by following the suggestions the paper has improved.

**Comment 2:**

(1) The section P7-L18-32 describes the effect of two additional functions (vegetation and build-structure) on dike stability. I found this description rather cryptic and unclear. It should be significantly improved. It is not clear, how either of the functions affects each breach mechanism. *[1]* Does the build structure affects only macro-instability due to additional weight? *[2]* Is there effect on piping, e.g. due to longer pipe length needed to induce a dike failure? *[3]* What do you mean by the insignificant amount of overtopping q<0.1 is acceptable? *[4]* What does this have to do with the structure or absence of a structure? *[5]* How one should imagine a scenario (with the annual probability of 1%), where a house disappears creating a hole at its location with the dike being intact (!). This is not clear to me.

**Response:**
We agree that this section can be made clearer and revised it based on this suggestion. The entire section has been revised to make it clearer. As the changes are extensive, please see the revised manuscript for the changes. The individual points are addressed here:

[1] The built structure is schematised by 2 effects in the model: 1) its additional weight and 2) a decrease in the maximum allowable overtopping rate.

[2] No, in our cases the structure does not extend into the aquifer and thus has little effect on piping. Furthermore one could argue the presence of a structure inhibits the well from emerging at the structure's location but a pipe could emerge just beside the structure instead along its outer wall.

[3] & [4] During overtopping the outer layer on the landward side of the dike should not erode during design conditions. When a structure is present it is effectively a discontinuity in the outer cover where water can more easily erode soil. This is reflected in a lower critical overtopping rate (q). When the structure is absent (see Table 1) there is presumed to be no grass cover but instead loose bare soil that will almost immediately start eroding. Since it is statistically impossible to rule out any overtopping (q=0 l/m/s) it is practise in the Netherlands to use q=0.1 l/m/s as a threshold value if effectively no erosion resistance can be expected and thus practically no overtopping is allowed during design conditions.

[5] The situation considered is not necessarily a collapse of the structure, but can also be when the structure is removed temporarily (e.g. a renovation or demolished intentionally to construct another). See also my response to reviewer 1. We will make sure to clarify this distinction in the revision.

**Comment 3:**

The second aspect is related to the first one and concerns the results of failure probability calculations (Fig. 5) and in particular the influence of different breach mechanisms. As the role of functions for various breach mechanisms was not clarified in details, it is very difficult to understand the effect of considering these functions on the distribution and changes of breach mechanisms presented in pie charts in Fig. 5. Unfortunately, the authors only scratch the surface and leave much of the presented results undiscussed and not analysed in-depth. I would appreciate a much more detailed analysis and discussion of the effects of (a) function failure and (b) reinforcement scenarios onto the role of breach mechanisms.

**Response:**
We agree with reviewer 2 and have explained this more in-depth by elaborating further on the reinforcements in section 3.4. Furthermore we expanded the results section to analyse the role of function failure for each of the reinforcement strategies as well as with an additional figure.

**Comment 4**

Finally, the authors mention in the text that uncertainties where somehow considered by considering scenario 7 "Robust dike". I did not understand this and do not see that uncertainties (of whatever nature) are considered here. Actually, the study is self-contained and there is no need to assess uncertainties as the probabilistic analysis already incorporates the uncertainties of various model parameters.

**Response:**
Indeed this sentence and its message were unclear. It was shown in the calculations that as the reliability of the dike increases, the influence of the uncertainty introduced by the functions has less influence on the failure probability. As the robust dike has the highest reliability, this effect was most clearly visible through this dike. The sentence has been revised to: "*The observation that the dike's own reliability influences the degree to which multifunctional use can affect the probability of failure of the dike was also found in this study.*"

**Comment 5:**

Abstract is poorly written and is not self-explaining. L11-14 are unclear for someone who has not read the paper and comes with general, though profound knowledge on flood risk.

**Response:**
Based on your and reviewer 1's comment the abstract has been rewritten. Please see revised manuscript.

**Comment 6:**

In overall, I rate this study as very solid and believe that after addressing the two major issues and a few minor comments below it can make an interesting and significant contribution to the research on probabilistic assessment of dike failures and flood risk assessment.

**Response:**
We are happy with the comments and have addressed the issues you have pointed out as best as we could.

*Minor issues*

**Comment 7:**

Introduction: The text is somewhat doggerel and needs a careful revision. (e,g, P1-L26-35 and comments below)

**Response:**

The section between P1-L26-35has been substantially shortened. Specifically by removing the section on the history of flood risk management in the Netherlands.

**Comment 8:**

P1-L19: risk of floods is not increasing everywhere. One should differentiate. "these cataastrophes" – you are talking of risk in general and not about some specific catastrophes.

**Response:**

While flood risks are in general increasing due to a combination of climate change (sea-level rise and extreme rainfall) and economic developments in deltas, I can imagine that in specific regions this is not the case. The sentence was revised to: *"many regions in the world are faced with increasing flood-risk."*

**Comment 9:**

P1-L21: "Risk based approaches have been" used not "performed". L22: remove "the" before "understanding".

**Response:**

Thank you for pointing out these corrections. They have been addressed.

**Comment 10:**

P1-L36-37: revise the sentence.

**Response:**

Upon more careful reading we concluded the sentence did not convey new relevant information and was therefore removed.

**Comment 11:**

P2-L1: Is this really true? The nation-wide risk assessment for England and Wales (Hall et al., 2003, 2005) also used probabilistic approach to assessment of protection level/failure probability.

**Response:**

While a probabilistic approach was certainly used there, by our knowledge it was not legally required to do so.

**Comment 12:**

P2-L35: Reference Hinkel et al. is missing in the reference list.

**Response:**

Hinkel et al. will be added to the reference list. We also manually checked each of the references and corrected other references where information was missing or incorrectly formatted.

**Comment 13:**

P3-L1: what is a 'cohesive' framework?

**Response:**

The word should have been coherent, not cohesive. This has been corrected.

**Comment 14:**

P3-L40 – P4-L1: as you mention, a conservative approach is usually taken assuming the NWO to be in the most critical state. Make clear that the actual probability of failure of the NWO is thus not considered.

**Response:**

The word "actual" has been inserted to make this clear*: "... because the actual probability of multifunctional elements being in a critical state is not considered".*

**Comment 15:**

P4-L9ff: in general, it seems that the vast majority of literature sources used in the manuscript is of Dutch origin. Nevertheless, there is also some relevant literature outside. E.g. the use of limit state functions and fault trees for flood defence assessment and hazard/risk assessment was performed by Kortenhaus (2003), Apel et al. (2004), Dawson & Hall (2006), Vorogushyn et al. (2009, 2010).

**Response:**

Indeed we could have used more international sources. The majority of references are of Dutch origin as the starting point of the research was the Dutch guidelines and sources/studies to support them. We have added Apel et al. (2004) and Vorogushyn et al. (2009) as references.

**Comment 16:**

P4-L28: what is WBI2017?

**Response:**

WBI2017 is the official abbreviation of the current Dutch assessment tools. References to WBI2017 were changed to "*the official Dutch assessment framework for flood defences*" . This particular sentence was removed as it repeated information presented in the introduction and aim.

**Comment 17:**

P5-L19: also Vorogushyn et al. (2009) compiled the statistics on dike failures from a few previous studies

**Response:**

Thank you again for the suggested literature. We have incorporated the suggestion in the manuscript.

**Comment 18:**

Eq.3.6: Use h=0 as the lower limit of the integral. –Inf dies not make sense for water levels.

**Response:**

If h would refer to water depth the lower limit would indeed make more sense to be 0. However since h can be negative in some reference systems (e.g. -1m +MSL which would be 1m below Mean Sea Level), h=-Inf is appropriate here. No change was made.

**Comment 19:**

P8-L3: Is this correct that the effect of the function is limited in the scenario 2? The yellow bar is significantly lower than for the monofunctional assessment! At P7-L40 you mentioned that in the scenario 2 there is a significant positive effect of the structure. Please, check.

**Response:**

As also pointed out by reviewer 1, this sentence should have referred to profiles 0,1 and **5** not 0,1 and **2**. It does not apply to 2 and 3 because here soil is replaced by additional weight of the structure leading to a net positive effect on stability. We have rewritten this section of the results in a clearer manner. Please see the revised manuscript.

**Comment 20:**

P9-L29-30: The sentence and the message is unclear to me.

**Response:**

The message was that before the new Water Act probabilistic assessments could be used but there was no obligation to do so. Now a probabilistic assessment is required and naturally more probabilistic assessments are being used. The sentence was revised to: "*Although probabilistic assessments have been used before, the new regulations of the Water Act in the Netherlands necessitate a full probabilistic assessment of flood defences.*"

**Comment 21:**

The list of references is not carefully formatted. Temmermann et al., journal missing.

**Response:**

The journal was added. Thank you for spotting this. We also manually checked each of the references and corrected other references where information was missing or incorrectly formatted.

**Comment 21:**

Move the equation for the Iribaren number from Table B1 into the B-section prior or after Eq. B10.

**Response:**

The equation was moved after Eq. B7.

**Changes made during revision**

| Page | From line | to line | Reason for revision | Change made |
|------|-----------|---------|---------------------|-------------|
| 1 | 8 | 18 | Both reviewers found the original abstract to be unclear and not self-explanatory. The abstract has therefore been entirely rewritten to better convey the message and concepts of the paper. | *See the submitted revision for the new abstract* |
| 1 | 21 | 22 | As pointed out by reviewer 2 the risk of floods is not increasing everywhere. | *From: "*With sea-level rising globally and an expected rise in extreme rainfall events due to climate change the risk of floods is increasing (Bouwer et al., 2010;Hirabayashi et al., 2013)."

 *To: "*With sea-level rising and an expected rise in extreme rainfall events due to climate change many regions in the world are faced with increasing flood-risk (Bouwer et al., 2010; Hirabayashi et al., 2013)" |
| 1 | 22 | | To make the introduction more concise as requested by reviewer 2, the introduction was shortened. This line did not convey new information and could therefore be omitted. | *Removed:* In order to develop sufficiently strong infrastructure to prevent flooding a framework is needed to assess the safety the infrastructure provides. |
| 1 | 23 | | Correction suggested by reviewer 2 | *Replaced:* "used" by the word "applied" |
| 1 | 24 | | Corrected an error | *Corrected:* "a better the understanding" to "a better understanding" |
| 1 | 27 | | To make the introduction more concise as requested by reviewer 2, the introduction was shortened by removing this section on the history of Dutch flood risk management | Removed:"Flood protection has always been a priority yet standards ... a full probabilistic approach (Delta Committee, 2008) and the change was made in 2017." |
| 1 | 28 | 37 | This section of the manuscript has been partially rewritten based on the suggestion of reviewer 2: "Introduction: The text is somewhat doggerel and needs a careful revision" | *See revised manuscript* |
| 2 | 2 | | Added the word "engineered" to clarify MFFDs are man-made structures | "Multifunctional flood defences (MFFDs) are engineered structures ..." |
| 2 | 5 | | Better word used | From: "... multiple additional ..." to: "... more ..." |

| 2 | 7 | | Reviewer 1 suggested: "the authors should revise the text if the context for the word function is always clear." We feel the use of the word is clear in this section. Still here this replacement is used to avoid repeating the word function too often. | From: "Other functions ..." to: "Multifunctional use of the flood defence ..." |
|---|---|---|---|---|
| 2 | 8 | | Simplified the sentence | From: "do not need to be a detriment to safety" to: "...does not need to decrease safety". |
| 2 | 9 | | The term "nature" was  in this context | From: "nature" to: "green foreshores" |
| 2 | 14 | | "Because they" fits better in the sentence | From: "... which ..." to: "... because they ..." |
| 2 | 16 | | No emphasis by the word especially was needed | Removed: "Especially" |
| 2 | 21 | | Shortened the sentence | From: "...to rules of thumb on the one hand and in-depth studies on the other." To:"... to rules of thumb and in-depth tailor-made studies." |
| 2 | 21 | 23 | Simplified and shortened this complex sentence | From: "Unless the multifunctional aspect is perceived to be of sufficient importance to justify a tailor-made study, assessments are often limited to showing other functions do not significantly diminish the safety of the flood defence while ignoring potential positive contributions to safety."

To: "Unless the multifunctionality is a key feature assessments are often limited to proving multifunctional use does not significantly diminish the safety of the flood defence ignoring potential positive contributions to safety." |
| 2 | 23 | 24 | Simplified and shortened this complex sentence | From: "Using a conservative approach for dike assessments where multifunctional elements can only negatively influence flood risk  does ensure safe dikes from a flood risk perspective but may hamper the implementation of efficient multifunctional dikes by requiring larger or more expensive dikes."

To: "Using such a conservative approach for dike assessments does ensure safe dikes from a flood risk perspective but may result in requiring larger and more expensive multifunctional dikes." |

| | | | | |
|---|---|---|---|---|
| 2 | 31 | 32 | Avoiding the term "functions" here as suggested by reviewer 1 | From: "to combine multiple functions with dikes" To: "for multifunctional use of the flood defence" |
| 2 | 34 | | Avoiding the term "functions" here as suggested by reviewer 1 | From: "functions" To: "multifunctional elements" |
| 2 | 37 | | Changed misused word as pointed out by reviewer 2 | From: "cohesive" To: "coherent" |
| 2-3 | 41 | 6 | Changed the sentences with an active structure using the word "we" into passive sentences. | *See revised manuscript* |
| 3 | 5 | | Monofunctional dikes were meant here as some traditional dikes were already multifunctional | From: "traditional dikes" to :" monofunctional dikes" |
| 3 | 9 | | Using a better word for the context | From: "communicated" to "documented" |
| 3 | 11 | | Using a better word for the context as pointed out by reviewer 1. See also the response to reviewer 1 comment 5 | From: "exclude" to "approve" |
| 3 | 20 | | Avoiding the term "function" here as suggested by reviewer 1 | From: "the function" to: "multifunctional use of the dike" |
| 3 | 24 | | Avoiding the term "functions" here as suggested by reviewer 1 | From: "functions" to: "multifunctional elements" |
| 3 | 34 | | Present tense fits better | From: "assessing" to "assess" |
| 3 | 35 | | Avoiding the term "functions" here as suggested by reviewer 1 | From: "functions" to: "multifunctional elements" |
| 3 | 35 | | Clarified the actual probability is calculated as suggested by reviewer 2 | From: "... the probability..." to: "... the actual probability ..." |
| 3 | 40 | | Added the word "mathematical" to avoid confusion with multifunctional use as suggested by reviewer 1 | "... mathematical limit state function ..." |
| 4 | 4 | 5 | Added 2 non-Dutch references as suggested by reviewer 2. | (Apel et al., 2004; ...; Vorogushyn et al., 2010) |
| 4 | 13 | | Avoiding the term "function" here as suggested by reviewer 1 | Changed: "function" To: "multifunctional element" |
| 4 | 18 | | This was later clarified by (Knoeff, 2017). This sentence can therefore be removed | Removed: "but it was left unclear how these failure probabilities can be implemented in the overall framework (Witteveen+Bos, 2013)." |
| 4 | 20 | | Introduce the approach earlier in the manuscript for clarity | Added: "This approach will be explored further in the study." |
| 4 | 23 | | Avoiding the term "functions" here as suggested by reviewer 1 | Changed: "functions" into: "multifunctional elements" |
| 4 | 23 | | Repeated information | Removed: "While in scientific ... evaluate multifunctional elements." |
| 4 | 23 | | Avoiding the term "functions" here as suggested by reviewer 1 | Changed: "functions" into: "multifunctional elements" |
| 4 | 23 | 24 | Simplified this sentence | From: "Through scenarios the inclusion of unspecified functions can be evaluated in different states through simple or complex models ..." |

| | | | | To: "Multifunctional elements can be evaluated in different scenarios with simple or complex models ..." |
|---|---|---|---|---|
| 4 | 25 | | Avoiding the term "functions" here as suggested by reviewer 1 | Changed: "function" into: "multifunctional element" |
| 4 | 25 | | Avoiding the term "functions" here as suggested by reviewer 1 | Changed: "function" into: "element" |
| 4 | 28 | | Avoiding the term "functions" here as suggested by reviewer 1 | Changed: "function" into: "multifunctional element" |
| 4 | 28 | 29 | Avoid using the word "we" as suggested by reviewer 1 | From: "We therefore synthesize the methods for MFFD assessments in the Netherlands in Fig. 2"

To: "Therefore the steps for MFFD assessments in the Netherlands are synthesized as follows (also see Fig. 2):" |
| 4 | 39 | | Replaced the word "approach" to "assessment" for consistency throughout the paper | From: "... difference between a basic approach and the risk-based approach ..."

To: "... difference between a basic assessment and a probabilistic one ..." |
| 4 | 39 | | Replaced the word "approach" to "assessment" for consistency within the paper | From: "In a basic approach ..."
To: "In a basic assessment ..." |
| 4 | 41 | | Typo corrected | From: "... found in steps 1 to 4 ..."
To: "... found in steps 1 to 5 ..." |
| 4 | 42 | | Replaced the word "risk approach" to "probabilistic assessment" for consistency within the paper | From: "... risk approach ..."
To: "... risk-based probabilistic assessment ..." |
| 4 | 42 | | Corrected with the proper tense | From: "... should be ..."
To: ".. are .." |
| 5 | 2 | | Replaced the word "approach" to "assessment" for consistency throughout the paper | From: "Comparing the basic framework with the expanded risk-based framework"

To: "Comparing the basic assessment with the expanded probabilistic assessment" |
| 5 | 3 | | Removed an unnecessary word | Removed: "more" |
| 5 | 4 | | Used a passive tense to avoid using the word "we" as suggested by reviewer 1 | From: "... we assess a set of MFFDs ..."
To: "...a set of MFFDs is assessed ..." |
| 5 | 4 | | Error corrected | From: "and a the traditional conservative approach"
To: "and a traditional conservative approach" |
| 5 | 5 | | Clarified table1 shows the approaches as suggested by reviewer 1. | From: "(see Table 1)."
To: "(see Table 1 for the approaches)." |

| 5 | 6 | | The concept of limit states has not yet been introduced in the paper. | From: "the limit state functions" To: "the models describing failure" |
|---|---|---|---|---|
| 5 | 7 | 8 | The concept of fragility curves has not yet been introduced in the paper. | From: "To combine the different failure probabilities the fragility curves of the mechanisms can be used (Bachmann et al., 2013) to arrive at the probability of failure." To: "The failure probabilities per scenario and failure mechanism are combined to arrive at the probability of failure." |
| 5 | 12 | | A low number can be better expressed with words | From: "3" to: "three" |
| 5 | 13 | | Added a reference as suggested by reviewer 2. | Added: Vorogushyn et al., 2009 as refference. |
| 5 | 16 | | Clarified a limit state function is a mathematical concept and not a form of multifunctional use as suggested by reviewer 1. | Added: "an equation called" |
| 5 | 23 | | Reference changed to the report detailing the experimental version of D-Stability used instead of the official release | From: "(Deltares, 2016)" To: "(Brinkman and Nuttall, 2018)" |
| 5 | 27 | 30 | The variables in the text have been rewritten with equation-tool to get the same formatting as in the equations. | *See revised manuscript* |
| 6 | 1 | | Year of publication was missing in the reference | Corrected to: (Hasofer and Lind, 1974). |
| 6 | 2 | | Better words used for the context | From: "find the low" to: "assess the small" |
| 6 | 10 | | Minor correction | Added the word relevant to: "all relevant water levels" |
| 6 | 22 | 23 | The sentence was better suited as an introduction to the section with some minor modifications | Moved and adapted sentence: From: "The dike is situated in ... a structure respectively" To: "The multifunctional dike for the case-study is situated in a riverine area, with nature on the floodplain side and a building on the landward side." |
| 6 | 24 | 25 | Made a list within the sentence for better readability | "with three methods: ..., ... and a ... ." |
| 6 | 27 | 32 | Section added explaining the effects of each reinforcement on the failure mechanisms as suggested by reviewer 2 | *See revised manuscript* |
| 7 | 1 | | Avoiding the term "function" here as suggested by reviewer 1 | From: "function" To: "multifunctional element" |
| 7 | 1 | | Clarification that only damage that results in dike failure needs to be considered. | From: "... can potentially damage part of the dike section." To: "... can compromise a section of the dike resulting in failure." |

| 7 | 2 | | Avoiding the term "function" here as suggested by reviewer 1 | From: "functions"
To: "multifunctional elements" |
|---|---|---|---|---|
| 7 | 3 | 5 | Revised the sentence and made a link to the appropriate profiles in Fig. 4. | From: "In alternatives 1 and 5 the use of the hinterland remains separate from the dike itself, while in the other alternatives the structure becomes an integral part of the flood defence."

To: "When broadening the dike on the flood plain or making a shallow outer slope (see profiles 1 and 5 in Fig. 4) the hinterland remains unaffected by the dike itself, while in the other alternatives the building becomes part of the flood defence" |
| 7 | 5 | | Making the sentence clearer | From: "...how the safety after the reinforcements is evaluated."
To: " ... the effect of the multifunctional elements on safety is evaluated." |
| 7 | 7 | | Avoiding the term "functions" here as suggested by reviewer 1 | From: "functions"
To: "multifunctional elements" |
| 7 | 8 | | Revised this sentence to be clearer | From: "The schematisation of functions in this study has been based on the fact-sheet by Knoeff (2017) ...."
To: "Effects of multifunctional elements on dike failure are incorporated through scenarios based on the fact-sheet by Knoeff (2017) ...." |
| 7 | 8 | | This is explained in the next sentences already | Removed: "for incorporating indirect failure mechanisms in assessments." |
| 7 | 9 | | The word element is more consistently used in the paper while it has the same meaning in this context. | From: "object" to: "element" |
| 7 | 13 | 36 | This section was entirely rewritten as suggested by reviewer 2. It was improved by including an explanation of how each function affects the different failure mechanisms, added a reference on the demolition of houses and a clarification on the "failed" state of a structure. | *See revised manuscript* |
| 7 | 25 | | Corrected the sentence by adding the word "the". | From: "... profile ..." to: "... the profile ..." |
| 7 | 29 | | Using the proper preposition | From: "... of the case study ..." to: "... in the case study ..." |
| 7 | 38 | | Textual error corrected | Removed: "... summarry ..." |
| 7 | 39 | 40 | Sentence was revised | From: "The probabilistic assessment of the functions and the monofunctional assessment yield a lower probability of failure." |

| | | | | To: "Both the probabilistic assessment of the additional multifunctional elements and the monofunctional assessment yield a lower probability of failure for each dike profile (Fig. 5)." |
|---|---|---|---|---|
| 7 | 40 | | Sentence contains no new information. | Removed: "Whether a function has a net positive or negative influence on the safety of the dike becomes only apparent by comparing these." |
| 8 | 1 | 31 | The entire section has been rewritten based on the suggestions of reviewer 2. Changes include:
• Headers for each failure mechanism
• Addition of Fig. 6 with fragility curves of each mechanism and profile in different states of the multifunctional components
• deeper analysis of the effects of the structure and its different states on the failure mechanisms
• the effect of the trees on the (piping) assessments | *See revised manuscript* |
| 9 | 5 | 6 | The meaning of this sentence was unclear | From: "... holds true for a conservative approach that only assesses the parts of the dike unaffected by other functions."

To: "... holds true for a conservative approach that omits multifunctional elements from the assessment." |
| 9 | 6 | | Avoiding the term "function" here as suggested by reviewer 1 | From: "... additional functions the ..."
To: " ... multifunctional elements their ..." |
| 9 | 7 | 8 | Avoiding the term "functions" here as suggested by reviewer 1 | Changed "functions" into "multifunctional elements" |
| 9 | 11 | | Avoiding the term "functions" here as suggested by reviewer 1 | Changed "functions" into "multifunctional elements" |
| 9 | 11 | | The word "failures" is clearer in this context | Changed : "risks" int "failures" |
| 9 | 16 | 17 | The exact meaning of "risks of functions" was not clear | From: "... new information on the risks of functions ..."
To: "...new information on the interaction between multifunctional uses and failure mechanisms ..." |
| 9 | 20 | | Avoiding the term "functions" here as suggested by reviewer 1 | Changed "functions" into "multifunctional elements" |
| 9 | 24 | | Avoiding the term "functions" here as suggested by reviewer 1 | Changed "functions" into "multifunctional elements" |
| 9 | 24 | 25 | Avoiding the term "functions" here as suggested by reviewer 1 | Changed "other functions" into "multifunctional use of the flood defence" |

| 9 | 25 | 27 | Sentence revised as suggested by the reviewers | From: "For piping Aguilar-López et al. (2015) demonstrated that reducing the uncertainty in  the seepage of  the soil of a multifunctional dike by correlating grain-size and hydraulic conductivity the probability of a piping failure is already reduced."

To: "For piping Aguilar-López et al. (2015) demonstrated that reducing the uncertainty in  the seepage properties of the soil of a multifunctional dike the probability of a piping failure is already significantly reduced." |
|---|----|----|----------------------------------------------|------------------------------------------------------------------------------------------------------------------------------------------------------------------------------------------------------------------------------------------------------------------------------------------------------------------------------------------------------------------------------------------------------------------------------------------------------------------------------------------------------------------------------------------|
| 9 | 28 |    | Replaced "ground" with "soil" for consistency with previous sentence | From: "ground" to "soil" |
| 9 | 29 | 31 | The sentence suggested uncertainties were only addressed by the robust dike (as pointed out by reviewer 2). | From: "The influence of uncertainties was also observed within this study through the case of a robust dike."

To: "The observation that the dike's own reliability influences the degree to which multifunctional use can affect the probability of failure of the dike was also found in this study." |
| 9 | 31 | 37 | Avoiding the term "function(s)" here as suggested by reviewer 1 | Changed "function(s)" into "multifunctional element(s)" or "multifunctional use of the flood defence" |
| 9 | 37 |    | Minor correction | Added "of" before "multifunctional use" |
| 9 | 39 |    | Avoiding the term "function" here as suggested by reviewer 1 | Changed "function" into "multifunctional use" |
| 9 | 39 |    | Revised the sentence | From: "...can have their own ..."
To: "...comes with its own ..." |
| 9 | 40 |    | Minor correction | Changed: "should" into "must" |
| 9 | 41 |    | Minor correction | Changed: "is" into "can be" |
| 9 | 42 |    | Added the word measures for clarity | From: "... flood protection" to "flood protection measures ..." |
| 9 | 43 |    | Minor correction | Changed: "should" into "need to" |
| 10 | 1 | 2 | Shortened the sentence to make it clearer | From: "While the current study looked at assessments for an existing situation, relating and managing uncertainties of functions to uncertainties in future climate conditions will be crucial for a probabilistic application of additional functions in designs."

To: "This study investigated the assessments of multifunctional flood defences for the current situation. In the design of these defences, however, future conditions, like for example |

| | | | | climate change or societal trends, need to be taken into account." |
|---|---|---|---|---|
| 10 | 2 | | Changed word to better fit the context | Changed: "Predictions" into "Scenarios" |
| 10 | 8 | | Changed "we" into "this study" as suggested by reviewer 1 | Changed "we" into "this study |
| 10 | 7 | 19 | Avoiding the term "function(s)" here as suggested by reviewer 1 | Changed "function(s)" into "multifunctional element(s)", "multifunctional use(s)" or "multifunctional use of the flood defence" |
| 10 | 10 | 11 | Revised this sentence as suggested by the reviewers | From: "Although a probabilistic assessment was not forbidden, new regulations and insights of the Water Act in the Netherlands stimulate a probabilistic assessment of flood protection."

To: "Although probabilistic assessments have been used before, the new regulations of the Water Act in the Netherlands necessitate a full probabilistic assessment of flood defences" |
| 10 | 11 | 12 | Specify that the framework is probabilistic | From: "a framework ... was synthesized"
To: "a probabilistic framework ... was developped" |
| 10 | 14 | | Specify that probabilistic assessment only always return lower assessed risks compared to conservative assessments | Added: "compared to conservative assessments" |
| 10 | 17 | 18 | Replaced the less clear term "protection level" with "reliability" | Replaced "protection levels" with "reliability" |
| 10 | 21 | 26 | Avoiding the term "function(s)" here as suggested by reviewer 1 | Changed "function(s)" into "multifunctional element(s)", "multifunctional use(s)" or "multifunctional use of the flood defence" |
| 10 | 24 | 25 | Added how scenarios are determined | Added: "These scenarios and associated probabilities will need to rely on expert judgment." |
| 10 | 25 | | Minor change | Changed: "Furthermore," into: "However," |
| 10 | 26 | | Combined the 2 paragraphs into 1 | - |
| 10 | 27 | | Not necessarily monitoring schemes need to be used to guide the scenarios of probabilistic assessments. | Removed "into monitoring schemes" |
| 10 | 27 | | Specified scenarios and their probabilities need further research | Added: "... on the proper scenarios and their associated probabilities ..." |
| 12 | 15 | | Reference to Bretschneider removed as the equations are | Removed (1957) |

| | | | | |
|---|---|---|---|---|
| | | | taken from another source based on Bretschneider. | |
| 12 | 21 | | Moved equation and updated the equation numbers accordingly | Moved equation $$\xi_0 = \frac{\tan(\alpha_{\text{out}})}{\sqrt{\dfrac{2\pi H_{\text{s}}}{gT_{\text{s}}^2}}}$$ out of table B1 |
| 13 | 5 | | Section on the ground water model has been omitted as this was only relevant for the heave and uplift sub-failure mechanisms of piping erosion. Because the internal erosion sub-mechanism of piping was dominant for all situations of this case-study, the uplift an heave sub-failures were eventually discared. Therefore the formulas and accosiated equations were obselete. The original Eq. (C3) stated the considered head difference for piping to be h*(1-r) which is wrong for the Sellmeijer formula. This equation only applies to the initiation of the heave and uplift sub-failure mechanisms where head differences under the blanket layer before resulting in an exit point are considered, not for the final internal erosion piping process described by Sellmeijer ultimately used in this study. | *Removed:* "Piping is evaluated with the ground water schematisation ... H=h*(1-r) (C3)" |
| 13 | 11 | 12 | WBI was not explained in the manuscript | From: "estimates provided for WBI assessments" To: "estimates used in Dutch dike assessments" |
| 13 | 15 | | WBI was not explained in the manuscript | From: "WBI 2017" To: "official Dutch guidelines" |
| 15 | 7 | | Revised the sentence to be clearer | From: "with the highest probability of occurring but rather converges to a local minimum" To: "with the highest probability but rather converges to a local design point" |
| 16-20 | - | - | References have been updated and missing information was added. Some references were removed or replaced (see below*) | *See further below* |
| 22 | Fig. 3 | | Caption corrected | From: "The probabilistic procedure for calculation the probability of failure of a dike cross-section in this study" |

| | | | | To: "The probabilistic procedure for calculating the probability of failure of a dike cross-section in this study" |
|---|---|---|---|---|
| 23 | Fig. 5 | | The piecharts for the probabilistic assessment and monofunctional assessment had been switched-up, this has been corrected.

Colors altered to work better when printing in grey-scale | *Figure has been updated* |
| 24 | Fig. 6 | | New figure to show the influence of the multifunctional elements on failure for every mechanism and profile | *Figure added* |
| 26 | Table 1 | | Avoiding the term "function(s)" here as suggested by reviewer 1 | Replaced the word "functions" with "multifunctional elements " |
| 26 | Table 1 | | Grammatical error corrected | Corrected: "... for a given failure mechanisms"
To: "...for a given failure mechanism" |
| 26 | Table 1 | | The words "of occurring" are not necessary | From: "a probability of occurring"
To: "a probability" |
| 26 | Table 1 | | Figure in the row probabilistic: X% changed to P for clarity based on reviewer 1's comment | *See manuscript* |
| 28 | Table B1 | | Formula for $\xi_0$ moved out of the table | *See manuscript* |

\* Additional changes to references

| Original reference | New reference | Reason |
|---|---|---|
| Kok, M., Jongejan, R., Nieuwjaar, M., and Tanczos, I.: Grondslagen voor hoogwaterbescherming, Ministerie van Infrastructuur en Milieu and Expertise Netwerk Waterveiligheid, Utrecht, the Netherlands, 2016. | Kok, M., Jongejan, R., Nieuwjaar, M., and Tanczos, I.: Fundamentals of Flood Protection, Ministry of Infrastructure and the Environment and Expertise Network for Flood Protection (ENW), Breda, the Netherlands, 2016. | Changed reference to the international version instead of the Dutch version |
| Slomp, R.: Flood risk and water management in the Netherlands : a 2012 update, Ministerie van Infrastructuur en Milieu and Rijkswaterstaat, Waterdienst, Utrecht, the Netherlands, WD0712RE205, 2012. | - | Omitted as the same information is present in the more easily accessible English book: "Fundamentals of flood protection" |
| Ministerie van Infrastructuur en Milieu: Wijziging van de Waterwet en enkele andere wetten (nieuwe normering | - | This has been removed as Ministerie van Infrastructuur en Milieu (2016b) contains the same information |

| | | |
|---|---|---|
| primaire waterkeringen). In: Kamerstuk 34 436 2016a. | | |
| Witteveen+Bos: Review notitie DHV/BomenwachtDT392-2, 2013. | - | Section with the reference was omitted |
| Delta Committee: Rapport Deltacommissie. Deel 1. Eindverslag en interimadviezen, 1960 | Maris, A.G., De Blocq van Kuffeler, V.J.P., Harmsen, W.J.H., Jansen, P.P., Nijhoff, G.P., Thijsse, J.T., Verloren van Themaat, R., de Vries, J.W., Van der Wal, L.T.: Rapport Deltacommissie. Deel 1. Eindverslag en interimadviezen, Delta Committee, Delft, the Netherlands, http://resolver.tudelft.nl/uuid:0e28dfd8-4e67-4267-a443-54b74a062bcb (last access: 21 february 2019), 1961. | Using the members of the Delta Committee as authors and added necessary information |
| Deltares: D-GEO STABILITY; Slope stability software for soft soil engineering, User manual, Deltares, The Netherlands, 2016. | Brinkman, R., and Nuttall, J. D.: Failure mechanisms – Macro Stability kernel; Scientific Background, Deltares, Delft, The Netherlands, 11201523-001-HYE-001, 2018. | Changed the reference from the manual of the official 2016 release to the 2018 report with the unofficial release used in the study |

[revised manuscript text omitted]

*parameters of the lognormal distribution based on (van Hoven, 2015)

[revised manuscript text omitted]

**Table C1 Description and values of variables in the piping limit state function**

| Variable | Description | Destribution | Parameters | Unit |
|---|---|---|---|---|
| $\gamma_{\text{p}}$ | specific weight of sand particles | Deterministic | 26 | $\dfrac{kN}{m^3}$ |
| $\gamma_{\text{w}}$ | specific weight of water | Deterministic | 10 | $\dfrac{kN}{m^3}$ |
| $\eta$ | drag factor | Deterministic | 0.25 | - |
| $\theta$ | bedding angle [°] | Deterministic | 35 | - |
| $RD$ $RD_m$ | Relative density of the material compared to small-scale piping experiments | Determinsistic | 1 | - |
| $d_{70m}$ | Reference d$_{70}$ of the material used in small-scale piping experiments | Determinsistic | $2 * 10^{-4}$ | $m$ |
| $m_{\text{p}}$ | Model factor for piping | Lognormal | $\mu = 1, \sigma = 0.12$ | - |